# Spatial defects nanoengineering for bipolar conductivity in MoS$_2$

Xiaorui Zheng[1,10], Annalisa Calò [1,2,10], Tengfei Cao[3,4,10], Xiangyu Liu[1], Zhujun Huang[1], Paul Masih Das [5], Marija Drndic [5], Edoardo Albisetti [1,6], Francesco Lavini [1], Tai-De Li[2], Vishal Narang[2], William P. King[7], John W. Harrold[8], Michele Vittadello[3,8], Carmela Aruta [9], Davood Shahrjerdi [1✉] & Elisa Riedo [1✉]

Understanding the atomistic origin of defects in two-dimensional transition metal dichalcogenides, their impact on the electronic properties, and how to control them is critical for future electronics and optoelectronics. Here, we demonstrate the integration of thermochemical scanning probe lithography (tc-SPL) with a flow-through reactive gas cell to achieve nanoscale control of defects in monolayer MoS$_2$. The tc-SPL produced defects can present either p- or n-type doping on demand, depending on the used gasses, allowing the realization of field effect transistors, and p-n junctions with precise sub-μm spatial control, and a rectification ratio of over 10$^4$. Doping and defects formation are elucidated by means of X-Ray photoelectron spectroscopy, scanning transmission electron microscopy, and density functional theory. We find that p-type doping in HCl/H$_2$O atmosphere is related to the rearrangement of sulfur atoms, and the formation of protruding covalent S-S bonds on the surface. Alternatively, local heating MoS$_2$ in N$_2$ produces n-character.

[1] Tandon School of Engineering, New York University, 6 MetroTech Center, New York, NY 11201, USA. [2] CUNY Graduate Center Advanced Science Research Center, 85 St Nicholas Terrace, New York, NY 10031, USA. [3] CUNY Graduate Center, Ph.D. Program in Physics and Chemistry, 365 5th Ave, New York, NY 10016, USA. [4] Department of Chemistry, College of Staten Island (CUNY), 2800 Victory Boulevard, New York, NY 10314, USA. [5] Department of Physics and Astronomy, University of Pennsylvania, Philadelphia, PA 19104, USA. [6] Dipartimento di Fisica, Politecnico di Milano, Via Giuseppe Colombo 81, Milano 20133, Italy. [7] Department of Mechanical Science and Engineering, University of Illinois, Urbana-Champaign, IL 61801, USA. [8] Department of Chemistry and Environmental Science, Medgar Evers College of CUNY, 2010, 1638 Bedford Ave, Brooklyn, NY 11225, USA. [9] National Research Council CNR-SPIN, University of Roma Tor Vergata, Viale del Politecnico 1, Rome I-00133, Italy. [10] These authors contributed equally: Xiaorui Zheng, Annalisa Calò, Tengfei Cao.
✉email: davood@nyu.edu; elisa.riedo@nyu.edu

Transition metal dichalcogenides (TMDCs) with formula $MX_2$ (M = metal and X = chalcogen) have emerged as an interesting class of two-dimensional (2D) materials, and have been employed as active components in various miniaturized electronic and optoelectronic devices[1]. Being able to control the 2D crystalline order and to spatially vary composition and structure[2] is of fundamental importance to tune their electronic properties. Examples include bandgap engineering in TMDCs alloys[3], substitutional doping at defects sites[4], spatial control of thickness[5] or of crystalline phases[6], and the growth of different laterally or vertically aligned TMDCs to form heterojunctions[7]. However, most of these methods are not scalable and it remains challenging to pattern with sub-μm resolution both n-type and p-type character within the same 2D material[8–10].

Various direct patterning methods have been demonstrated in literature, by using localized electric fields, electron radiation, or laser writing. Scanning tunneling microscopy (STM) has been used to characterize single defects in monolayer 2D materials, and to induce defects by local electrochemical reactions[11–16]. Oxidation-scanning probe lithography was recently applied to pattern-insulating barriers on MoS$_2$ flakes[17]. Electron-beam radiation is also known to generate chalcogen vacancies in 2D materials[18,19]. Recently, an electron-beam-induced technique has been demonstrated to p- and n-dope encapsulated graphene and MoS$_2$ heterostructures[20]. Finally, laser writing has been used for local oxidation, thinning, or patterning of monolayer TMDCs[21–23].

Defects engineering has been proposed as a viable way to induce n-character in MoS$_2$, in particular sulfur vacancies (S-vacancies) can produce n-type conduction in MoS$_2$[10]. However, aside from S-vacancies, very little is known about other types of defects in monolayer MoS$_2$, including their atomistic characteristics, their formation mechanisms, and their impact on electronic properties[14,24]. A comprehensive understanding of defects would also be beneficial to find strategies to heal their detrimental effects[25], and to postpone defect-induced sample degradation over time[26].

Here, we propose an approach to control defects and doping in monolayer MoS$_2$ which is scalable, does not require alignment marks, and allows for both n-type and p-type doping. By integrating thermochemical scanning probe lithography (tc-SPL)[27–34] with a flow-through reactive-gas cell, we spatially and thermally activate different types of defects in 2D MoS$_2$, presenting either p- or n-type doping on demand with nanoscale resolution. These patterned defective regions are clearly observed in Kelvin Probe Force Microscopy (KPFM). Specifically, when the hot tc-SPL nanoprobe heats the surface of MoS$_2$ in HCl/H$_2$O atmosphere, it produces local p-type doping, whereas local tc-SPL heating in an inert N$_2$ atmosphere produces n-type doping. The electronic properties of the patterned areas are then investigated by using tc-SPL for the complete fabrication of field-effect transistors (FET) and channel doping. The nature and structure of the here-formed defects are studied at the atomic level by scanning transmission electron microscopy (STEM), X-ray photoelectron spectroscopy (XPS), and density-functional theory (DFT). The p-character of locally heated MoS$_2$ in HCl/H$_2$O atmosphere is found to be related to the rearrangement of sulfur (S) atoms and the formation of new protruding covalent S–S bonds on the surface, whereas local heating of MoS$_2$ in N$_2$ atmosphere produces S-vacancies and consequently n-type character. Finally, an electronic p–n junction in monolayer MoS$_2$ has been realized by tc-SPL p-type and n-type doping with precise spatial control, demonstrating a state-of-the-art current rectifying capability, with a rectification ratio of over $10^4$. This approach, being based on direct writing with tc-SPL, offers unique features including in situ imaging of monolayer MoS$_2$, no need of physical masks and alignment marks, nanoscale-patterning resolution, and potential for scalability.

## Results

**HCl/H$_2$O–tc-SPL of p-type defects.** Mechanical exfoliation is considered the best method to produce single-crystal TMDCs, such as MoS$_2$, with the highest structural quality and environmental stability[26] for high-performing electronic devices[35]. On the other hand, chemical vapor deposition (CVD) allows producing extended TMDCs monolayers with acceptable crystallinity[36], and tens of micrometer size at controlled positions on substrates[37]. In general, exfoliated mono- and multilayer MoS$_2$ contain intrinsic defects, such as S-vacancies[38,39], missing MoS$_2$ layers, charge traps, and scattering centers. The most common intrinsic point defects in CVD MoS$_2$ are S-vacancies[37], because of their low activation energy, ~2 eV[40], while Mo vacancies alone are unlikely to exist[41]. This fact is reflected in the CVD MoS$_2$ stoichiometry, where the S/Mo ratio appears to be often lower than 2[42]. S-vacancies can be also generated in MoS$_2$ during post-exfoliation or post-growth processes, for example by thermal treatments[43], plasma exposure[44], sample characterization and imaging[45], or environmental exposure[26].

In this work, we thermally induce different types of localized defects in exfoliated and CVD monolayer and multilayer MoS$_2$ by integrating tc-SPL with a flow-through gas cell, which allows for environmental control during tc-SPL patterning. Initially, through an inert N$_2$ stream, vapors from a diluted HCl solution are flowed inside the gas cell where an exfoliated MoS$_2$ sample is located (Fig. 1a). The gas cell contains HCl/H$_2$O vapors and N$_2$ from the inert gas carrier. High-temperature scans are performed by tc-SPL on different areas of the MoS$_2$ flakes at different scan rates, while keeping a constant N$_2$ flow, HCl concentration, and heater temperature. The induced defects in the patterned regions are investigated by KPFM and Raman spectroscopy (Fig. 1). XPS, DFT, and STEM are used to understand the atomistic structure of these defects, as explained in the following paragraphs.

HCl/H$_2$O–tc-SPL patterns and the unheated region of the same flakes are characterized by KPFM (Fig. 1b). We observe a strong contrast between the tc-SPL patterns and the nonpatterned regions in KPFM, where the tc-SPL heated areas always exhibit a lower contact potential difference (CPD) value (top inset of Fig. 1b). The tc-SPL pattern, a rectangular area of $1.2 \times 0.3$ μm$^2$, shows a CPD 80 mV lower compared with the surrounding unheated flake. The thickness fluctuations after the tc-SPL treatment are identified to be ~0.1 nm, corresponding to <5% of the total thickness of the used three-layer MoS$_2$ flake (top inset of Fig. 1b). The corresponding AFM topographical image of the pristine sample is also shown in the bottom inset of Fig. 1b. The observed CPD contrast can be interpreted as a Fermi energy[46] shift toward the valence band, which locally increases the work function (Φ) of the HCl/H$_2$O–tc-SPL heated areas compared with the nonpatterned areas. In Fig. 1c, we report $\Delta\Phi_{KPFM}$ ($= \Phi_{pattern} - \Phi_{non\text{-}patterned}$) vs. the inverse scan rate ($1/\nu$). $\Delta\Phi_{KPFM}$ increases (more p-type) with increasing $1/\nu$ (decreasing scan rate), consistent with a thermally activated Arrhenius process[34]. In Fig. 1c, the data are fitted according to a first-order kinetic reaction equation developed for the case of a hot tip sliding on a thermally reactive polymer surface[47] (Supplementary Note 1 and Supplementary Fig. 1). Fitting is performed assuming that the local thermochemical reaction, i.e., the production of defects with doping nature, is proportional to the observed $\Delta\Phi$, according to the following equation[48]:

$$\Delta\Phi = \Delta\Phi_{pattern} - \Delta\Phi_{non-pattern} = \Delta\Phi_0 \cdot \left( 1 - e^{-\frac{A \cdot r}{\nu} \cdot e^{-\frac{E_a}{RT}}} \right), \quad (1)$$

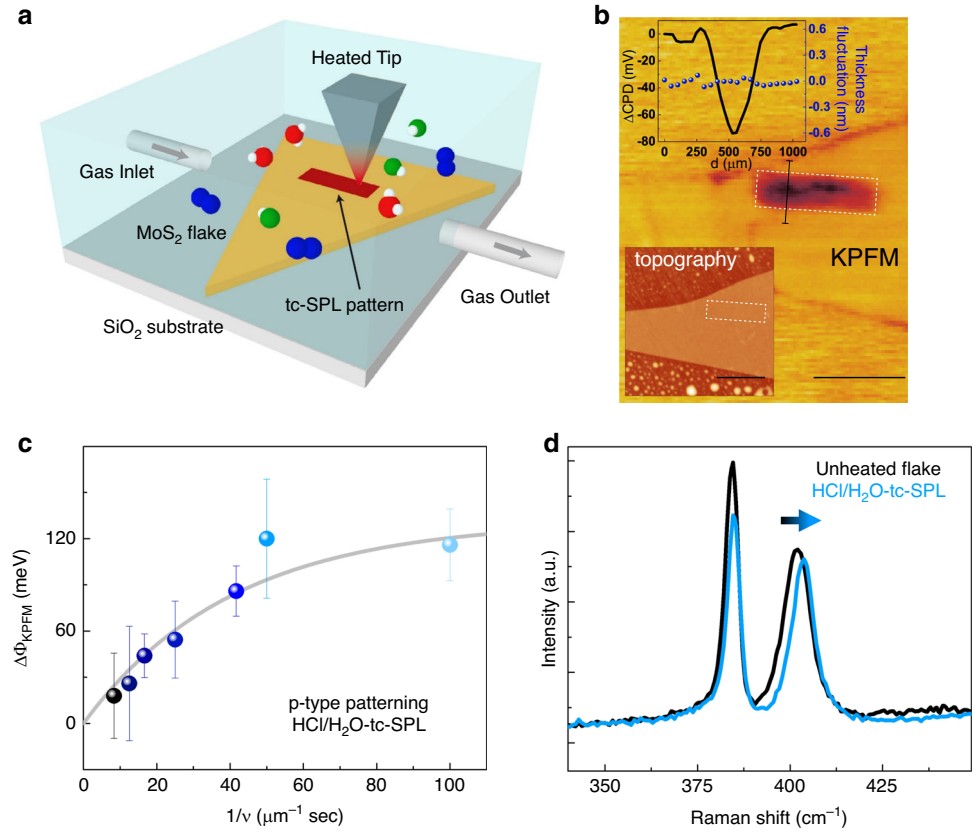

**Fig. 1 tc-SPL in HCl/H₂O atmosphere. a** Schematics of the tc-SPL experiment. **b** KPFM image of a three-layer exfoliated MoS₂ flake, where a pattern of 1.2 × 0.3 µm² has been produced at a high temperature ($T = 873$ K) and at a scan rate of $\nu = 0.024$ µm s⁻¹ in a HCl/H₂O-rich environment (scale bar: 1 µm, z scale: 160 mV). This pattern is highlighted by white boxes in the image. Top inset shows the cross-section profile of the KPFM image across the pattern as indicated with a solid line in (**b**), specifically we plot both the contact potential difference (ΔCPD = CPD_pattern−CPD_non-patterned) and the thickness fluctuation. Bottom inset shows the AFM topography of the same pristine flake (scale bar: 1 µm, z scale: 25 nm). **c** Work-function difference between the tc-SPL pattern and the nonpatterned MoS₂ flake $\Delta\Phi_{KPFM} = (\Delta\Phi_{pattern} - \Delta\Phi_{non-pattern})$ vs. inverse patterning scan rate, and the corresponding data fit with Eq. (1) (shown as a gray line in (**c**)). The error bar is the standard deviation of KPFM data within the area of analysis. **d** Raman spectra collected inside (data in light blue) and outside (data in black) a tc-SPL heated pattern in single-layer exfoliated MoS₂.

where $\Delta\Phi_0$ is the maximum work-function difference, and $A$, $r$, $E_a$, $R$, and $T$ are, respectively, the Arrhenius constant, the tip-surface diameter, the activation energy, the gas constant, and the absolute temperature[47]. For a fixed temperature, $T = 823$ K, and $R = 70$ nm, the fitting procedure gives $A = 10^{12}$ s⁻¹, $E_a = 2.03$ eV, and $\Delta\Phi_0 = 131.7$ meV (Supplementary Fig. 2). The obtained value of $A$ is in agreement with the transition-state theory[48], where $A$ is given by $k_B T/h$ ($k_B$ and $h$ are the Boltzmann constant and Planck constant, respectively). This frequency represents the number of collisions with the correct molecular orientation of converting the reactants to the product and should be $\sim 10^{12}$ s⁻¹ at room temperature. Furthermore, the obtained value of $E_a$ is similar to the formation energy of S-vacancies[40], which are involved in the rearrangement of S on the surface and formation of S–S bonds giving p-character to the tc-SPL-activated areas, as explained later by DFT calculations (see below). Figure 1d shows the Raman spectra inside (data in light blue) and outside (data in black) a HCl/H₂O–tc-SPL pattern on an exfoliated monolayer MoS₂ flake. The energy difference between the two Raman modes ($A_{1g} - E^1_{2g}$) is in agreement with the monolayer nature of the flake ($\sim 19$ cm⁻¹)[49]. A blue shift of $\sim 2$ cm⁻¹ is observed for the $A_{1g}$ peak in the tc-SPL-heated region, while the $E^1_{2g}$ peaks remains approximately the same position. Previously, blue/red shifts of the $A_{1g}$ peak have been observed in case of chemical doping of MoS₂ and are attributed to p/n doping, respectively[43,50]. Therefore, the here observed blue shift of the $A_{1g}$ Raman peak

indicates local p-character in the HCl/H₂O–tc-SPL-patterned MoS₂ samples and is consistent with the work-function increase found in KPFM. The smallest pattern we achieved by tc-SPL (width: 70 nm) is shown in Supplementary Fig. 3.

**Mechanism of HCl/H₂O–heat activated p-type defects.** The mechanism behind the HCl/H₂O–tc-SPL p-type defects in MoS₂ is elucidated by XPS, DFT calculations, and STEM imaging. Because the minimal XPS beam spot size is 200 µm, in order to extract meaningful information, XPS experiments have been conducted on samples macroscopically heated in the same atmospheric conditions used in the tc-SPL experiments (see "Methods"). As shown in Fig. 2, the stoichiometry of exfoliated MoS₂ is obtained through XPS elemental analysis for globally heated exfoliated MoS₂ samples in pristine conditions (data in black) and after heating at 573 K (data in blue) and 673 K (data in cyan) in a HCl/H₂O-rich environment. XPS elemental analysis of the Cl 2p core levels indicates the absence of chlorine (Cl) in all MoS₂ samples treated in HCl/H₂O atmosphere. The S/Mo total area ratio (Fig. 2a), obtained from the fit of the Mo 3d and S 2p core-level spectra, is slightly higher than 2 for the pristine and heated samples, indicating that exfoliated MoS₂ is a sulfur-rich (or molybdenum-deficient) material. The valence band (VB) spectra (Fig. 2b) show a shift toward the lower binding energies with increasing temperature. The same shift is observed for both Mo 3d (Fig. 2c) and S 2p (Fig. 2d) core-level spectra, indicating

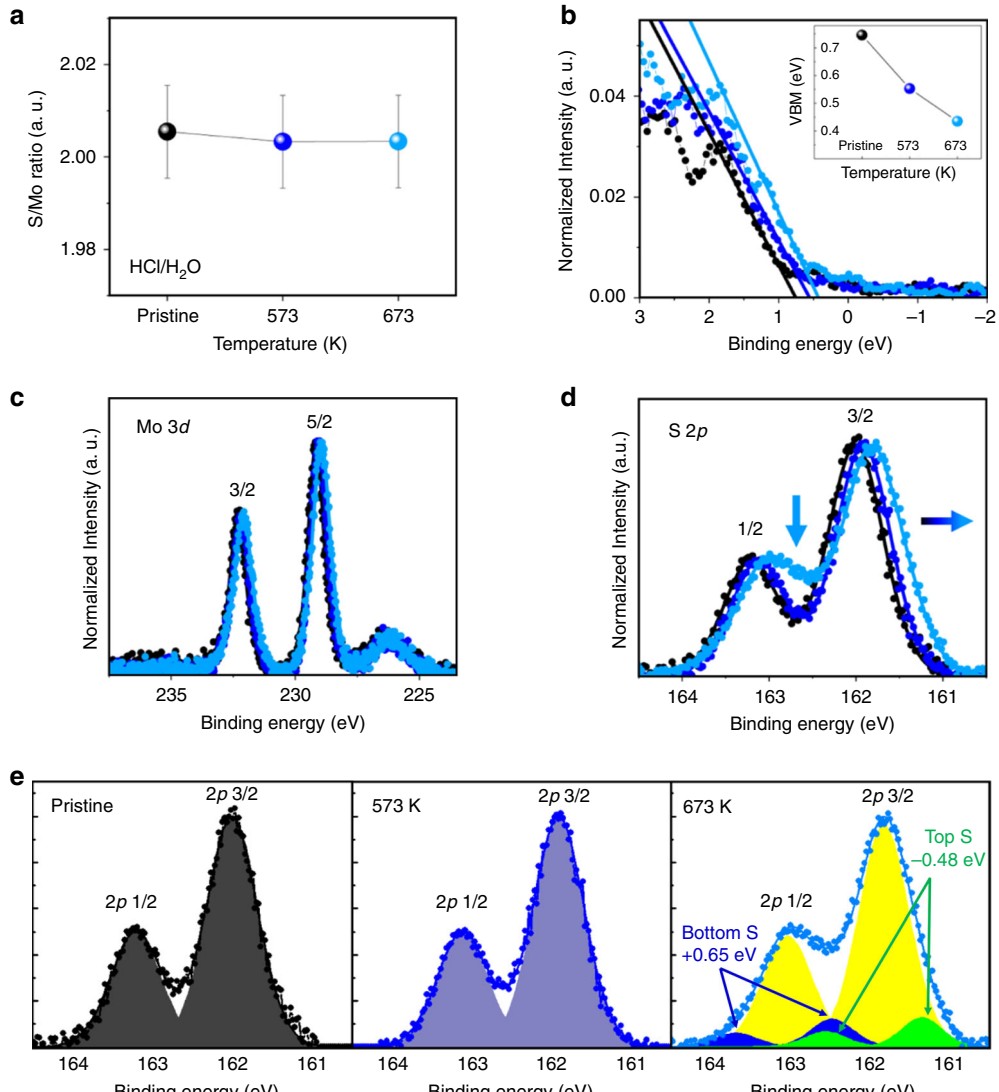

**Fig. 2 XPS of multilayer exfoliated MoS₂ heated in HCl/H₂O atmosphere. a** Stoichiometry of exfoliated multilayer MoS$_2$ samples in pristine conditions (data in black) and heated at two temperatures (573 K, data in blue and 673 K, data in cyan) in HCl/H$_2$O atmosphere. The error bar is determined via different spectra measured for each condition. **b** Valence band (VB) spectra (normalized to area) close to the Fermi edge for the same samples. The inset shows that the valence band maximum (VBM), as measured from the Fermi level, shifts toward the Fermi edge with increasing temperature. VBM is obtained from the intersection with zero of the linear fit of the leading edge region in the VB spectra. **c** Mo 3$d$ and **d** S 2$p$ core-level spectra for the three samples, from which the S/Mo ratio is obtained. The horizontal arrow in (**d**) indicates the direction of the S 2$p$ core-level shift toward lower binding energy with increasing p-character and temperature, while the vertical arrow indicates the region where the S 2$p$ spectrum at 673 K is different from those of the pristine material and exfoliated MoS$_2$ heated at 573 K. **e** Fit of S 2$p$ core levels for pristine sample (left), sample heated at 573 K (middle), sample heated at 673 K (right), where two additional doublets are used for the fitting.

that the whole XPS spectra undergo a rigid shift toward the Fermi level (zero binding energy) at high temperatures (the corresponding survey scans of the valence band of MoS$_2$ and of the SiO$_2$ substrate are shown in Supplementary Fig. 4). The valence band maximum (VBM, inset of Fig. 2b), extrapolated from the leading edge of the VB (Fig. 2b), shifts toward lower binding energy with increasing temperature. Since in XPS the binding energies are obtained with respect to the Fermi level, the decreased energy difference between the VBM and the Fermi level could be related to a Fermi level shift toward the VB[51], see Supplementary Note 2 and Supplementary Fig. 5. This finding is in agreement with the observed work-function increase in KPFM (Fig. 1b, c) due to an induced p-type character in MoS$_2$ when heated in HCl/H$_2$O atmosphere. However, the S/Mo ratio does not significantly change with the incubation temperature (see the

blue and cyan data points in Fig. 2a), indicating that the main reason for the induced p-type character cannot be the sample stoichiometry variation[14].

Moreover, the S 2$p$ core-level spectra appear clearly broadened after heating at 673 K (Fig. 2d, see also Supplementary Fig. 6). Indeed, while the S 2$p$ core levels of the pristine and the 573 K heated samples are fitted using a single 2$p_{3/2}$ and 2$p_{1/2}$ spin–orbit doublet, the fitting of the S 2$p$ spectra of the 673-K heated sample requires additional spin–orbit doublets (Fig. 2e). As discussed later in the DFT section, an excellent fitting of the 673 K S 2$p$ peak has been obtained with three spin–orbit doublets (Fig. 2e). The additional components, related to covalent S–S bonds[52], have a relative area of ~8.9% of the total spectrum area. The fitting shows that 8.9% of sulfur atoms exhibit a higher binding energy shift (+0.65 eV), while 8.9% of sulfur atoms have a lower binding

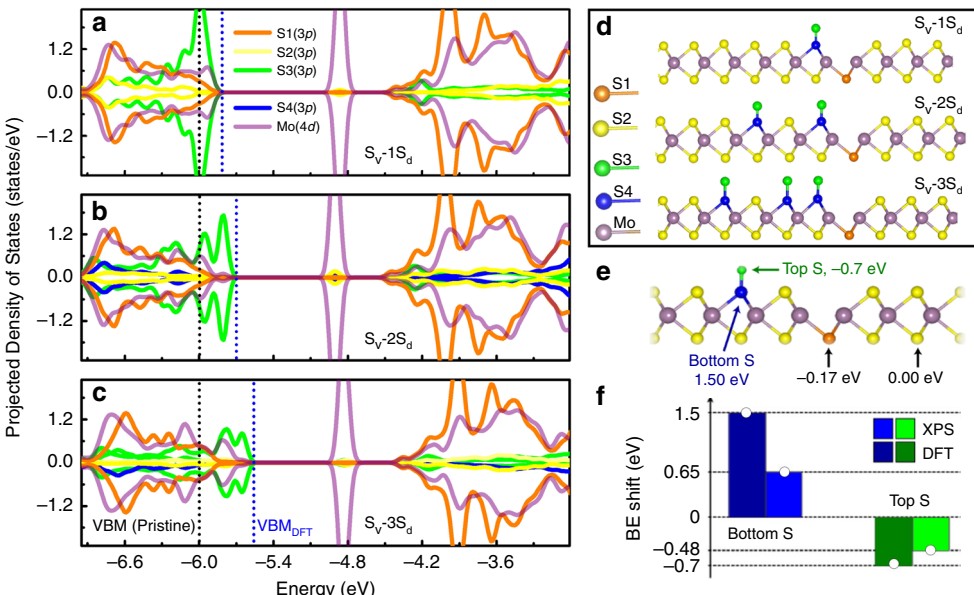

**Fig. 3 DFT calculations.** Atom normalized spin-polarized projected density of states (PDOS) of $MoS_2$ with one S vacancy and one ($S_v$-$1S_d$) (**a**), two ($S_v$-$2S_d$) (**b**), and three ($S_v$-$3S_d$) (**c**) protruding surface S atoms chemisorbed on the surface. The energy level of the valence band maximum ($VBM_{DFT}$) is calculated with respect to the vacuum level, and it is indicated by a blue dashed line, while the VBM of pristine $MoS_2$ is shown with a black dashed line and obtained from Ref [37]. **d** The schematic figures of the corresponding $MoS_2$ structures used in PDOS calculations. The S atom (S1) underneath a S vacancy, the regular S atom (S2), and the protruding S atom (S3) covalently bonded to a S atom on the surface (S4) are colored in orange, yellow, green, and blue, respectively. Mo atoms are represented by violet balls. **e** The core-level energy shifts of S $2p$ states with respect to regular S atom (S2) (0.00 eV) are indicated. **f** Histogram of the DFT results (in dark blue and dark green) for the bottom (S4) and protruding S (S3) atoms with creation of a S vacancy together with the XPS energy shifts (in light green and light blue) of the two additional doublets used for the fit.

energy shift (−0.48 eV) with respect to the regular sulfur atoms in the $MoS_2$ lattice. This indicates that the chemical bonds in exfoliated $MoS_2$ are modified after heating in $HCl/H_2O$ atmosphere, and the induced p-type character of $HCl/H_2O$-tc-SPL-treated $MoS_2$ is clearly related to this modified chemical bonds, which will be discussed more in detail with the aid of DFT calculations and STEM measurements. We propose that both the XPS core-level energy shifts and the resulting p-type character are due to the formation of surface-protruding S–S covalent bonds, occurring through a redistribution of adsorbed S atoms on the $MoS_2$ surface, and/or through the relocation of S atoms on the $MoS_2$ surface after vacancies formation. Therefore, two structural models are adopted here, namely, rearrangement of already existing extra S atoms on the surface of $MoS_2$, and/or redistribution of new S atoms from S-vacancies creation, with both scenarios producing covalent S–S bonds without changing significantly the sample stoichiometry. These two scenarios are also corroborated by the XPS observation that pristine exfoliated $MoS_2$ samples have a S/Mo ratio slightly higher than 2 (Fig. 2a), and there is no signature of other foreign elements. We remark that Mo vacancies can hardly form in $MoS_2$[41].

The DFT calculations on surface-protruding S–S covalent bond models demonstrate that top sites of S atom in the $MoS_2$ matrix are the most stable adsorption-positions of extra S atoms (Supplementary Fig. 7). The projected density of states (PDOS) of $MoS_2$ with one S vacancy and an increasing number of protruding surface S atoms are calculated and shown in Fig. 3a (one protruding surface S atom, $S_v$–$1S_d$), Fig. 3b (two protruding surface S atoms, $S_v$–$2S_d$), and Fig. 3c (three protruding surface S atoms, $S_v$–$3S_d$), where the energy levels of the states are calculated with respect to the vacuum level. The corresponding $MoS_2$ structures used in PDOS calculations are shown in Fig. 3d. It is evident that the protruding surface S atoms are the major contributors to the $VBM_{DFT}$ shift toward higher energy levels with respect to pristine $MoS_2$ (see also Supplementary Fig. 8). The

shift of $VBM_{DFT}$, calculated from the vacuum level, has been correlated with XPS and KPFM results, and a good agreement has been achieved (Supplementary Note 2). Moreover, we demonstrate that Mo vacancies do not produce p-type doping in $MoS_2$ (Supplementary Fig. 9). Considering also their large formation energy, we exclude that Mo vacancies are present in our experiments.

Core-level binding energy of S atoms in $MoS_2$ is calculated by DFT and used as a reference to obtain the energy shifts of other possible S states on the $MoS_2$ surface, such as S–S protruding bonds (Supplementary Fig. 10). It is observed that there is a large variation on the core-level energy shift of S atoms under different lattice environments. The process shown in Fig. 3e, where a surface S atom chemisorbs on the top site of $MoS_2$ and creates a stable S–S bonding configuration by leaving behind a vacancy in monolayer $MoS_2$, is in agreement with the almost constant S/Mo ratio observed by XPS (Fig. 2a) when heating at different temperatures in $HCl/H_2O$ atmosphere. The projected DOS of monolayer $MoS_2$ shows that the $3p$ states of S–S bonding atoms induce a p-type character in $MoS_2$ (Supplementary Fig. 11). When considering S atoms in the bottom- and top-site positions within the protruding S–S bonds, DFT calculations give positive (+1.5 eV) and negative (−0.7 eV) binding energy shift, respectively (Fig. 3e). These shifts are in agreement with those observed in XPS, namely, a positive shift of +0.65 eV, and a negative shift of −0.48 eV, for the two additional doublet components used to fit the XPS 673 K S $2p$ peak (Fig. 2e). We notice that DFT also predicts a shift of −0.17 eV for the S atom below the S vacancy with respect to the regular S atom in the perfect $MoS_2$ monolayer crystal (Fig. 3e). However, such binding energy shift is smaller than the full width at half-maximum (FWHM) of the main doublet peaks, which therefore include both the main lattice and the below-vacancy S atoms. It should be noted that in our DFT calculations, various models have been proposed and calculated for defective $MoS_2$. The shifts obtained by XPS and DFT are

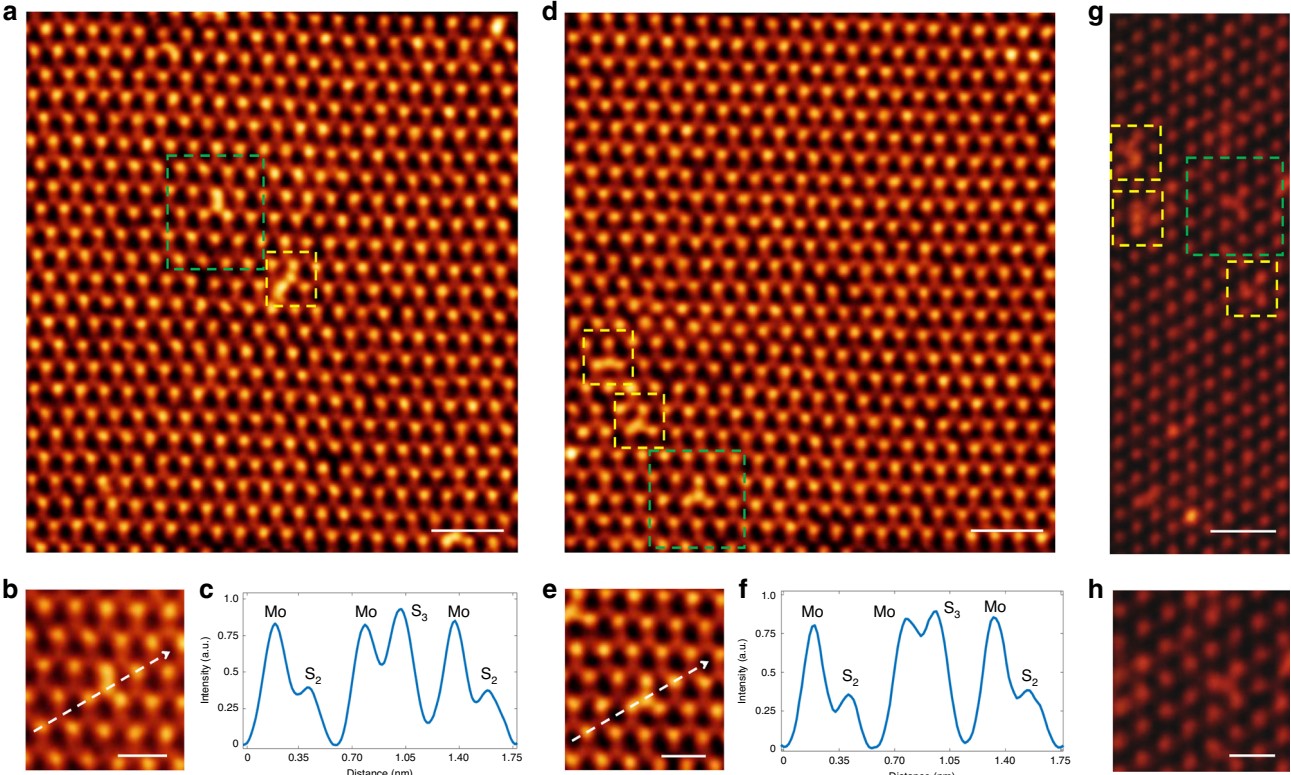

**Fig. 4 STEM characterization of HCl/H$_2$O p-type defects in MoS$_2$. a** HAADF STEM lattice image of a CVD-grown monolayer MoS$_2$ flake heated in HCl/H$_2$O for p-character treatment, as described in the main text. As a result of the mass contrast behavior (~$Z^2$, where $Z$ = atomic number) of HAADF imaging, Mo atoms ($Z$ = 42) appear as bright spots in a 2H (trigonal prismatic) coordination with S$_2$ ($Z$ = 32) lattice sites. We observe the presence of dopants (outlined in green and yellow) that are not seen in pristine, untreated samples. **b** High-magnification STEM image of an individual dopant from (**a**) (outlined in green), in which a noticeable increase in contrast is detected at a chalcogen lattice site. **c** Intensity profile across the dopant site reveals an intensity rise that is consistent with an additional protruding S atom, the S$_3$ dopant. **d**-**f** Numerous dopants observed across different CVD MoS$_2$ flakes demonstrate similar structural and contrast features. **g**, **h** HAADF STEM lattice image of an exfoliated monolayer MoS$_2$ flake exposed to p-character treatment, showing S$_3$ dopants. Scar bar: 1 nm in (**a**), (**d**), and (**g**); 500 pm in (**b**), (**e**), and (**h**).

compared in Fig. 3f. While the agreement between the DFT and XPS shifts is qualitatively good, it has some limitations due to the restrictions of this comparison. For example, XPS measurements have been performed on multilayer MoS$_2$, since flakes with different numbers of layers are present within the beam radius, while DFT calculations have been performed on a monolayer. Thus, we expect some differences between the ideal MoS$_2$ monolayers simulated in DFT, and the experimental exfoliated multilayers structures.

High-angle annular dark-field (HAADF) STEM images of MoS$_2$ samples are collected to reveal the atomic structure of the defects formed when heating MoS$_2$ in HCl/H$_2$O atmosphere. The atomic resolution STEM images of individual defects in monolayer CVD and exfoliated MoS$_2$ are shown in Fig. 4. The mass contrast behavior of HAADF imaging, where the contrast is proportional to $Z^2$ with $Z$ = atomic number, reveals a noticeable intensity rise at a chalcogen lattice site, which cannot be seen in the untreated pristine samples. This increased intensity indicates the presence of an additional S atom on top of a surface S atom in MoS$_2$ (this type of defect is hereafter called S$_3$ dopant), confirming a rearrangement of S atoms and the formation of protruding covalent S–S bonds on the surface (see model in Fig. 3e). By averaging over multiple STEM images, we obtain a S$_3$ defect density of roughly 2–4% of the total chalcogen sites, which corresponds to ~$2.4 \times 10^{13}$ to $4.7 \times 10^{13}$ defects·cm$^{-2}$.

**N$_2$–tc-SPL of n-type defects**. The tc-SPL method can also achieve or increase the n-type character of MoS$_2$. Indeed, when tc-SPL experiments are performed in an inert environment, i.e., flowing only N$_2$ inside the gas cell, the tc-SPL patterns always show higher CPD (smaller work function) compared with the unheated regions of the same MoS$_2$ flakes. This is the opposite KPFM contrast as compared with that observed for the HCl/H$_2$O–tc-SPL doping. To understand the evolution of the work-function variation in N$_2$-tc-SPL doping, we investigate $\Delta\Phi_{\text{KPFM}}$ for CVD monolayer MoS$_2$ as a function of the absolute tc-SPL heater temperature (Fig. 5a, for scan rate = 0.2 µm s$^{-1}$), which is then fitted according to Eq. (1). At low temperatures, the work function inside the locally heated areas is similar to the nonpatterned flake ($\Delta\Phi_{\text{KPFM}} \approx 0$). Starting from T$_{\text{Heater}}$ = 900–1000 K, the work function starts to decrease significantly and stabilizes around −60 meV. We note that the effective temperature at the tip–sample contact is lower than the heater temperature (Supplementary Note 1)[31]. This $\Delta\Phi_{\text{KPFM}}$ evolution is consistent with a thermally activated process[34,47]. The fitting analysis gives a maximum change in work function of $\Delta\Phi_0$ = −57.9 meV, an activation energy of $E_a$ = 2.40 eV, and Arrhenius constant of $A$ = $10^{12}$ s$^{-1}$, when using as fixed fitting parameters $R$ = 70 nm, and $\nu$ = 0.2 µm s$^{-1}$. We remark that the values of activation energy and collision frequency $A$ are similar to the ones found in the HCl/H$_2$O case, and consistent with S-vacancy formation and transition-state theory, respectively.

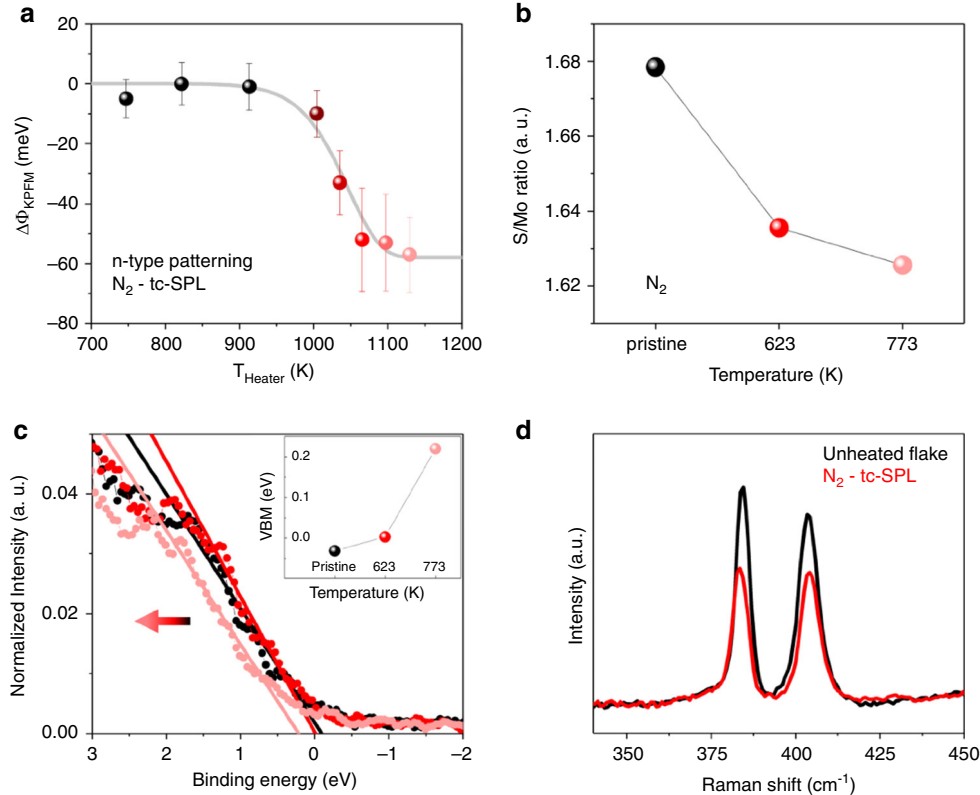

**Fig. 5 tc-SPL in $N_2$ atmosphere. a** Work-function difference $\Delta\Phi_{KPFM} = \Phi_{pattern} - \Phi_{non-patterned}$ vs. absolute temperature and corresponding data fit with Eq. (1) (shown as a gray line). The error bar is standard deviation of KPFM data within the area of analysis. **b** Stoichiometry of single-layer CVD $MoS_2$ in pristine conditions (data in black) and heated at two temperatures (623 K, data in red and 773 K, data in pink) in $N_2$ atmosphere, as obtained by XPS core levels spectra. **c** XPS valence band spectra (normalized to area) close to the Fermi edge for the same samples. The horizontal arrow indicates the direction of the shift toward higher binding energy with increasing n-character and temperature. Inset shows the valence band maximum evolution as a function of the temperature. **d** Raman spectra collected inside (data in red) and outside (data in black) a tc-SPL pattern in single-layer CVD $MoS_2$.

**XPS of $N_2$–heat-activated n-type defects**. In order to explain the temperature-induced decrease of work function in $N_2$ atmosphere, we perform XPS on globally heated CVD $MoS_2$ samples (Fig. 5b, c) and Raman spectroscopy (Fig. 5d) inside and outside locally heated $N_2$–tc-SPL patterns. Elemental analysis from XPS (Fig. 5b) shows that the S/Mo ratio in monocrystalline single-layer CVD $MoS_2$ is lower than 2 and decreases with the temperature. This decrease is associated with S-vacancies, which increase for the samples heated at higher temperatures[14] ($T = 623$ K and 773 K). The XPS VB spectra and the VBM values as measured with respect to the Fermi level show a shift toward higher binding energies (the Fermi level moves away from the VB) with the increase of temperature (Fig. 5c), in agreement with the decreased $\Delta\Phi$ measured by KPFM (Fig. 5a). This indicates a higher n-type character induced by local heating in $N_2$ atmosphere. Raman spectra collected inside (data in red) and outside (data in black) a $2 \times 2$ μm$^2$ tc-SPL-patterned area in $N_2$ atmosphere (Fig. 5d) also confirm the presence of n-type defects in $N_2$–tc-SPL patterns. In this case, we do not observe any shift of the $A_{1g}$ peak, while the $E^1_{2g}$ peak shows a slight red shift of ~0.7 cm$^{-1}$. Furthermore, both $A_{1g}$ and $E^1_{2g}$ peaks have lower intensities compared with the unheated flake. Red shift of the $E^1_{2g}$ Raman peak accompanied by lower peak intensity and peak widening of the $A_{1g}$ and $E^1_{2g}$ peaks have been observed in electron irradiated $MoS_2$ samples and attributed to the formation of S-vacancies and lattice deformation[45,50]. We also notice that the extracted energy difference between $A_{1g}$ and $E^1_{2g}$ peaks is ~20 cm$^{-1}$, as expected for a monolayer CVD sample[53].

XPS elemental analysis also shows an increase in the content of oxygen with the temperature (Supplementary Fig. 12), which may indicate a thermally induced substitution of S by oxygen[43]. While oxygen incorporation in 2D $MoS_2$ has been linked to p-type conduction[54], S-vacancies in $MoS_2$ have been identified as a source of n-doping in $MoS_2$ by several studies[10]. In particular, S-vacancies have been shown to introduce defects in the $MoS_2$ bandgap[55], a fact that pins the Fermi energy closer to the conduction band[56] and makes $MoS_2$ a n-type semiconductor[42]. However, recently, two independent STM investigations revealed a slow oxygen-substitution reaction, during which individual sulfur atoms are replaced one by one by oxygen, giving rise to a solid-solution-type 2D $MoS_{2-x}O_x$[11,57]. One of these studies showed that this process, obtained either in air at room temperature, or at 400 K, gives rise to $MoS_{2-x}O_x$ samples with n-character. However, they have not been able to determine whether this n-character was due to the oxygen substitutions, since the same n-character was already present in the pristine $MoS_2$ samples, therefore reaching no clear conclusions[11,57]. In particular, the authors of Ref. [57] indicate that substitutional O can be incorporated in $MoS_2$ also while annealing in vacuum, because previously adsorbed oxygen molecules on vacancy sites could split and leave the O behind. Our $MoS_2$ samples are flowed with $N_2$ before starting the tc-SPL experiments, and the surface is imaged by AFM in contact mode, likely removing $H_2O$, $O_2$, or $CO_2$ adsorbed on the surface. During the tc-SPL process, the cell is filled with $N_2$, and we therefore argue that we create S-vacancies, producing n-type doping in the sample. However, we remark that prolonged exposure of the patterned/doped samples in air

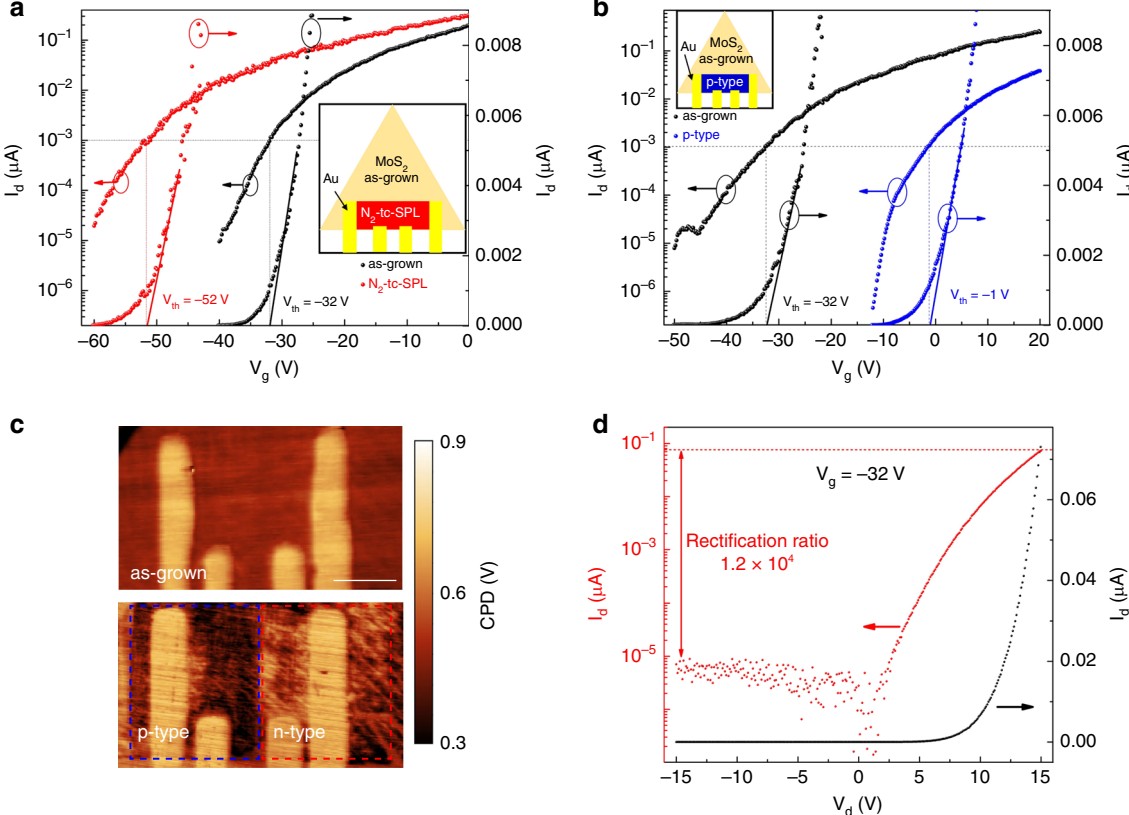

**Fig. 6 Electrical characterization of tc-SPL doped FETs. a** Transfer curves of a four-probe FET before (black curves) and after (red curves) N₂-tc-SPL n-type doping of the full channel. Inset: schematic figure showing the FET with the whole channel being n-type doped by N₂-tc-SPL. **b** Transfer curves of a four-probe FET before (black curves) and after (blue curves) tc-SPL p-type doping of the full channel. Inset: schematic figure showing the FET with the whole channel being p-type doped by tc-SPL. **c** KPFM images of a FET channel before (top panel) and after (bottom panel) the formation of the lateral p–n junction. Both the p-type half-channel (blue dashed box) and the n-type half-channel (red dashed box) are realized by using tc-SPL method. Scale bar: 2 µm. **d** Output curve of the lateral p–n junction in logarithmic (left axis) and linear scale (right axis), obtained at the optimal back gating of −32 V. A rectification ratio >10⁴ has been achieved.

produces unstable levels of n-type doping, possibly due to the formation of oxygen substitutional defects[11]. Furthermore, n-type doping is dramatically influenced by the state of the as-grown MoS₂ sample, e.g., by the number of vacancies present at the time of the doping procedure, which are strongly dependent on the local humidity. On the other hand, tc-SPL p-type doping is more reproducible and stable (see "Methods").

**Electrical characterization of tc-SPL-doped FETs**. We fabricate three different types of FETs, namely, a FET with a fully n-type doped channel by N₂-tc-SPL, a FET with a fully p-type doped channel by HCl/H₂O-tc-SPL, and a FET in which half-channel is p-type doped and the other half is n-type doped, obtaining a lateral p–n junction. The electrodes of these FETs are also fabricated by using thermal probes, as previously reported[58] (see "Methods"). The electrical characterization of these FETs is performed before and after tc-SPL doping.

After fabrication of a four-probe FET on a monolayer CVD MoS₂, we dope the entire active area/channel by using tc-SPL in N₂ (see inset of Fig. 6a). The transfer curves before and after n-type doping are plotted in Fig. 6a in black (as-grown) and red (N₂-tc-SPL), respectively. A threshold voltage of −32 V is extracted for the as-grown FET before N₂-tc-SPL n-type doping, by using both the constant current method in logarithmic scale (left axis), and the linear extrapolation method in linear scale (right axis)[59]. After N₂-tc-SPL doping of the active region, the

threshold voltage shifts from −32 V to −52 V, confirming an induced n-type doping.

Next, we perform KPFM measurements on this FET before and after the n-type doping. From these measurements, we obtain a work-function variation ($\Delta\Phi_{KPFM} = \Phi_{n-type} - \Phi_{as-grown}$) of −18 ± 5 meV, in agreement with the data reported in Fig. 5a, when the tc-SPL heater temperature is 1000 K. Here, we define "±" representing the standard deviation on KPFM data within the area of analysis out of five repetitions of KPFM measurements. Note that the substrate is grounded during the KPFM measurements.

To correlate the electrical FET measurements with the KPFM results, we use the following equation for approximating the work-function variation ($\Delta\Phi_{FET}$) of a non-degenerate semiconductor as a function of its carrier density[60]:

$$\Delta\Phi_{FET} = \Phi_{n-type} - \Phi_{as-grown} = \ln\left(\frac{n_{as-grown}}{n_{n-type}}\right), \quad (2)$$

where $\Phi_{as-grown}$ and $\Phi_{n-type}$ are the work functions of the as-grown and n-doped MoS₂, respectively; $n_{as-grown}$ and $n_{n-type}$ are the gate-bias-dependent carrier densities of the as-grown and n-doped MoS₂, extracted from the FET electrical measurements. In particular, we approximate the bias-dependent carrier density in the channel region of the MoS₂ FET in strong inversion (i.e., at

gate bias beyond the threshold voltage) using[61]:

$$n = C_{ox}\left(V_g - V_{th}\right), \qquad (3)$$

where $C_{ox}$ is the gate capacitance, $V_g$ is the back-gate voltage, and $V_{th}$ is the threshold voltage. From Eq. (2) and using the bias-dependent carrier density before and after the n-doping (Supplementary Fig. 13), we estimate a work-function shift ($\Delta\Phi_{FET}$) of $-12$ meV at $V_g = 0$ V (Supplementary Fig. 14). This result is consistent with the KPFM data.

To study the electrical properties of p-type doped $MoS_2$ by $HCl/H_2$-tc-SPL, we fabricate a second type of four-probe FET on a monolayer CVD $MoS_2$. We dope the entire active area/channel using $HCl/H_2$-tc-SPL (inset of Fig. 6b). The transfer curves before and after p-type doping are plotted in Fig. 6b in black (as-grown) and blue (p-type), respectively. The threshold voltage shifts from $-32$ V (as-grown) to $-1$ V after $HCl/H_2$-tc-SPL p-type doping. This positive shift of $V_{th}$ is consistent with the p-type doping of $MoS_2$.

We then study the work-function shift in the channel region of the $MoS_2$ device before and after the p-doping. We use the same analysis applied to the n-type devices earlier. In particular, the work function change obtained from KPFM is $\Delta\Phi_{KPFM} = \Phi_{p\text{-type}} - \Phi_{as\text{-grown}} = +90 \pm 6$ meV. In comparison, the work-function change estimated from the FET data at $V_g = 0$ V is $\Delta\Phi_{FET} = \Phi_{p\text{-type}} - \Phi_{as\text{-grown}} = +90$ meV (Supplementary Figs. 15, 16), in good agreement with the KPFM data.

Finally, we fabricate a lateral p–n junction by doping the channel with the tc-SPL method. We start by n-doping the entire channel of a FET using $N_2$-tc-SPL, and then convert the doping type in one half of the channel region to p-doping by $HCl/H_2$-tc-SPL. Figure 6c shows the KPFM image of the final device, indicating the formation of a lateral p–n junction. By using the KPFM image of as-grown $MoS_2$ as a reference (Fig. 6c top), it is possible to identify the p-doped and n-doped regions of the lateral p–n junction (Fig. 6c bottom), where the darker CPD area (marked with a blue dashed box) corresponds to the p-doped region and the brighter CPD area (marked in red dashed box) corresponds to the n-doped region.

The output characteristic curves of the FET after the initial n-type doping of the entire channel confirms the absence of a diode-like characteristic behavior for all the measured gate voltages (Supplementary Figs. 17, 18). This is evident from the symmetric output characteristic curves of the FET. After half-channel $HCl/H_2$-tc-SPL p-doping, however, the device shows a gate-bias-dependent rectification behavior. In particular, the device achieves a maximum rectification ratio, defined as the ratio of forward current (ON) over reverse current (OFF) of the p–n junction, over $10^4$ at $V_g = -32$ V (Fig. 6d). The rectification ratio decreases when further increasing the gate bias, approaching 1 (Supplementary Fig. 18). The transition from a diode-like behavior at low gate bias to a unipolar device characteristic curve at high gate biases (Supplementary Figs. 17, 18) is due to the gradual increase in the density of gate-bias-induced electron charge carriers relative to the tc-SPL-induced p-type carriers. These gate-bias-dependent characteristic curves are consistent with previous reports on lateral p–n junctions[62]. The strong rectification behavior of the lateral tc-SPL fabricated p–n junction provides further evidence of the ability of tc-SPL in altering the doping type and doping level at precise locations of a $MoS_2$ device on demand.

## Discussion

We show that tc-SPL patterning is a strategy for the design with precise size and spatial control of electronic p–n junctions in monolayer $MoS_2$, where a current rectifying ratio of over $10^4$ is observed. We remark that environmental tc-SPL can be employed for $MoS_2$ patterning in optoelectronics, flexible electronics, and energy-harvesting applications, which require complementary devices with regions of different polarity. This has been particularly challenging to realize in $MoS_2$ because of its natural propensity for unipolar n-type transport[42,44]. Here, the rectifying junction is produced by direct imaging and writing in a one-step fabrication approach, without the necessity of additional fabrication steps for markers, mask deposition, or resist development. Our approach and understanding of thermally activated defects formation in $MoS_2$ constitute a way to introduce functionalities in 2D materials through defects nanoengineering, and could find wide applications in semiconducting TMDCs, for example to engineer metal–2D semiconductor electrical contacts, reducing the Schottky barrier height, and as a doping strategy for electronic and optoelectronic devices.

Finally, it is important to underline that here we have demonstrated that the entire FET can be fabricated by tc-SPL, from electrodes patterning[58] to bipolar doping, offering features which include in situ imaging of monolayer $MoS_2$, no need of physical masks or markers, nanoscale-patterning resolution, and potential for scalability. Current tc-SPL throughput is similar to EBL[29], but it can be further improved by multiplexing with thermal nano-probes arrays[28,63], paving the way for integrated 2D electronic devices.

## Methods

**Materials.** To test different types of $MoS_2$ materials, we use either mechanically exfoliated monolayer and few-layer (1–3 layers) $MoS_2$, or monocrystalline monolayer $MoS_2$ grown by CVD (from 2d layer). All samples are deposited on a 285-nm-thick $SiO_2$ film on highly doped silicon substrates (from Nova Electronic Materials). Flakes thickness is confirmed by Raman spectroscopy, AFM, and optical microscopy before experiments.

**tc-SPL nanopatterning.** In tc-SPL[27–33] heatable conical nano-probes (radius ~35 nm) are used to induce chemical reactions on the surface of thermally responsive materials and thin films with nanoscale resolution. Highly doped silicon cantilevers with an integrated heater, i.e., a region at lower doping levels located at the tip position, are used to control the thermal writing parameters. tc-SPL heaters are switched-on through a DC voltage, and the current flowing in the highly doped cantilever legs resistively raises the tip temperature, as high as 800–1000 °C. Writing is achieved by scanning the tc-SPL hot nanoprobe on the sample surface. Here, we integrate tc-SPL with a flow-through reactive-gas cell. In particular, to produce the $HCl/H_2O$ atmosphere during the tc-SPL process, a solution of HCl is prepared in milli-Q water at a concentration of 2.4 N starting from a 10 N solution (Fisher). A sealed glass flask containing 50 mL of the solution is connected through plastic tubes to a $N_2$ gas line and to the inlet of the tc-SPL gas cell, where the sample is placed. The reactive gas is sealed in the gas cell between the sample holder and the scanning head of an atomic force microscope (AFM, 5500 model from Keysight), which has been designed to hold the cantilevers with integrated heater for tc-SPL. Another tube is connected to the outlet of the gas cell to a hood for vapor exhaust. In this way, $N_2$ flows from the main line through the HCl solution carrying $HCl/H_2O$ vapors inside the AFM gas cell. The HCl gas from a 2.4 M solution was passed through a NaOH solution for 20 min. The change in pH was monitored to calculate the HCl gas concentration. We obtained the 74 micromoles per liter of gas. To quantify the HCl gas concentration in the gas cell, HCl from a 2.4 M solution was passed through a NaOH solution for 20 min. The change in pH was monitored to calculate the HCl gas concentration, obtaining 74 micromoles·$L^{-1}$. Before tc-SPL patterning, AFM images are collected in contact mode with unheated tc-SPL probes in order to find the appropriate flake. Then, $N_2$ is passed through the HCl solution at a flow rate of 60 mL $min^{-1}$, which is compatible with good images and patterning quality and preserves the flakes integrity. The $N_2$ flow is also controlled during the measurements by recording the rate of bubbles formation when the gas enters the $HCl/H_2O$ solution (~6 bubbles $s^{-1}$). We observe that very high $N_2$ flow lead to flakes disruption and produces noise in the AFM images. In Fig. 1c, different patterns are produced on each flake at different scan rates from 0.02 to 0.12 $\mu m\ s^{-1}$, keeping constant the other experimental parameters, i.e., probe temperature, gas flow, and load. Calibration of the tip temperature during the experiments is performed before starting the $N_2$ flow and far from contact[64]. In the case of tc-SPL patterning in pure $N_2$ atmosphere, $N_2$ is directly flown inside the gas cell at a rate of 40 mL $min^{-1}$ for 30 min, while performing images and patterning of the flake in contact mode. Then the tc-SPL experiment is performed under $N_2$ flow, by scanning equally spaced $1 \times 0.2\ \mu m^2$

rectangular areas of the same flake at increasing temperature and constant load, and scan rate = 0.2 μm s$^{-1}$.

**tc-SPL parameter optimization.** To achieve tc-SPL doping, a wide range of parameters have been studied and optimized in order to achieve the most reproducible doping of $MoS_2$. These parameters include: (1) the concentration of the HCl solution; (2) the gas flow rate; (3) the gas pre-flow duration; (4) the set-point load of the thermal cantilever; (5) the number of scanned lines along the slow axis ($y$ axis); (6) the scan rate; and (7) the temperature of the heater. We have performed over 500 p-type doping experiments to evaluate the influence of the seven parameters on the HCl/$H_2O$-tc-SPL doping of $MoS_2$ in terms of doping level, uniformity, doping speed, flake damage, and reproducibility. After the optimization, we have fabricated in total 30 p-type doping devices and observed consistent work-function change for 28 of these devices (yield ~93%). It should be noted that the successful n- and p-doping level of the devices depends also on the type of $MoS_2$ samples, e.g., exfoliated flakes vs. CVD flakes. Regarding n-type doping, the results are less reproducible because they strongly depend on the original samples, and how the samples are stored. Figures 1c and 5a present the results of some of the p- and n-doping experiments, where the doping level is measured as a function of scan rate and temperature after all the other parameters have been optimized.

**KPFM measurements.** Room temperature amplitude and frequency modulation KPFM (AM-KPFM and FM-KPFM) is performed using a Bruker Multimode AFM (Santa Barbara, CA), with PPP-EFM tips (Nanosensors). Lift scan height of 20 nm and ac bias voltage of 2 V are used during measurements. The sample is grounded during the measurements. For data analysis, contact potential difference values (average and standard deviation) are extracted from same size areas inside and outside the tc-SPL patterns for different samples.

**Raman measurements.** Raman spectra are collected using a Horiba LabRAM HR800 system coupled with an Olympus BX41 inverted optical microscope, and using a laser source with excitation wavelength of 532 nm. The laser power is adjusted to avoid sample damage, or any sample modification, as observed by optical microscopy. Spectra are acquired between 100–900 cm$^{-1}$ with 1 s exposure time and as an average of ten different measurements. The peak at 521 cm$^{-1}$ from the silicon substrate is used as a reference.

**XPS measurements.** For the XPS characterization, mechanically exfoliated a few-layer-thick $MoS_2$ samples and single-layer CVD $MoS_2$ samples with crystal size of the order of 50–100 μm are globally heated at different temperatures in different atmospheric conditions. For heating exfoliated samples in HCl/$H_2O$ atmosphere, a hot plate is used. A small Büchner flask (total volume: 25 mL) with a gas inlet is put upside down on top of exfoliated $MoS_2$ samples lying on an aluminum foil, and then the foil is wrapped around the flask. $N_2$ gas is flown through an aqueous HCl solution (2.4 N, 50 mL) inside the flask for 30 min at room temperature (RT), and the rate of bubbles formation inside the HCl solution is checked to be equal to that occurring during tc-SPL experiments (~6 bubbles s$^{-1}$). Then the flask with the sample is transferred to the hot plate, keeping the $N_2$ flow. Two samples are heated, one at 573 K for 5 min, and the other one at 673 K for 10 min. The actual temperature on the hot plate is checked using a thermocouple. We use a pristine $MoS_2$ sample, a sample kept for 190 min in HCl/$H_2O$ atmosphere at room temperature, and a sample heated in humid air (RH = 50%) at 673 K for 10 min as control samples in the XPS measurements (Supplementary Figs. 6, 12). For such samples, we do not find any difference in the XPS S $2p$ core-level spectra of $MoS_2$ compared with the pristine sample. To mimic tc-SPL experiments in atmosphere of pure $N_2$, CVD $MoS_2$ samples are put in a furnace filled with nitrogen gas ($N_2$ flow = 10 SPLM). Temperature is raised in 20 s to the set-point value and then kept constant for 10 min. Then it is reduced back to the RT value in 300 s. Two experiments are performed using different set-point temperatures (623 K and 773 K). No changes are observed by optical microscopy after thermal treatments on both exfoliated and CVD samples. XPS spectra are collected using a Versaprobe II XPS (Physical Electronics) using Al Kα radiation (1486.6 eV). We collect Mo $3d$, S $2p$ core-level spectra and valence band (VB) spectra using the following conditions: pass energy = 23.5 eV (Mo $3d$ and S $2p$) and 46.95 eV (VB), beam size = 200 μm, time per step = 200 s, resolution = 0.025 eV step$^{-1}$. Samples are mounted on a steel sample holder, and grounded and measurements are performed at a pressure <10$^{-6}$ Pa. Binding energy calibration is performed with reference to the Si $2p$ peak from the $SiO_2$. For the fitting, a Shirley function is assumed for background subtraction and we use a multicomponent deconvolution procedure, using mixed Gaussian and Lorentzian line shapes. For the Mo $3d$, we use the $3d_{5/2}$ and $3d_{3/2}$ spin–orbit splitting fixed at 3.14 eV and the degeneracy ratio at 2:3 for the spin–orbit area ratio. For the S $2p$, we use the $2p_{3/2}$ and $2p_{1/2}$ spin–orbit splitting fixed at 1.2 eV and the degeneracy ratio is 1:2 for the spin–orbit area ratio. The FWHM is a free parameter, but is kept as the same value for the components of each doublet. The S/Mo stoichiometry ratio is obtained from the Mo $3d$ and S $2p$ peak areas weighted with the relative sensitivity factors (9.5 and 1.67 for Mo $3d$ and S $2p$ core levels, respectively).

**DFT calculations.** DFT calculations are carried out with the QUANTUM-Espresso software[65]. In particular, norm-conserving pseudopotentials[66] are adopted to describe electron–ion interactions, and the Perdew–Burke–Ernzerho (PBE) functional[67] is used to give electron–electron exchange and correlation effects. The plane wave energy cutoff is set to 100 Ry to get accurate charge density, and with these DFT schemes, the lattice constant of $MoS_2$ is 3.255 Å in agreement with experiments[68]. Large rectangular monolayer $MoS_2$ are roots to analyze possible existing states of sulfur clusters on $MoS_2$ and corresponding electronic structures. In all simulations, 16.27 Å × 16.91 Å supercells are applied to exclude the interaction between defects with their periodic images, and a 15-Å vacuum size along the $z$ direction is applied to remove $MoS_2$ layer–layer interplay with its mirror duplicates. The corresponding Brillouin zone is sampled by a 2 × 2 × 1 mesh using the Monkhorst–Pack scheme[69] in the supercell optimizations. For DOS calculations, a large 4 × 4 × 1 mesh is used. All the structures are fully relaxed within the force threshold of 1 × 10$^{-3}$ Ry bohr$^{-1}$. The Methfessel–Paxton smearing technique[70] with the width of 0.02 Ry is applied to speed up the convergence. The core-excited pseudopotential methods are used to calculate core-level binding energies of $2p$ states of S atoms. Norm-conserving pseudopotentials are generated for photo-excited ions, including a hole in the $2p$ subshell of S atoms. These pseudopotentials are derived from scalar-relativistic all electron density-functional calculations, in which spin–orbit coupling terms are averaged within each state. Therefore, our calculated S $2p$ core-level binding energies correspond to an average binding energy for the entity.

**STEM measurements.** HAADF STEM images of $MoS_2$ samples were collected at Lehigh University to confirm the formation of protruding covalent S–S bonds on the surface of $MoS_2$ when heating $MoS_2$ in HCl/$H_2O$ atmosphere. Monolayer CVD-grown flakes (2d-layer supplies) and multilayer exfoliated flakes from bulk $MoS_2$ crystals (SPI supplies) on Si/$SiO_2$ were used. Doping was performed under $N_2$ previously flown through HCl solution (2.4 N) at exactly the same condition as for the XPS measurements. STEM measurements were obtained using a spherical aberration-corrected JEOL 200ARM-CF with an acceleration voltage of 80 kV. Images were taken with a high-angle annular dark-field (HAADF) detector with a detection range of 54–220 mrad and 10-cm camera length while electron radiation damage was minimized by using a low electron probe current (11 pA).

**FET fabrication and characterization.** All FETs are fabricated by thermal scanning probe lithography (t-SPL) method, which gives rise to high-quality metal contacts with vanishing Schottky barriers. First, a solution of pure PMGI (poly-methylglutarimide, Sigma) is spin-coated on the samples surface (2000 r.p.m., 35 s) followed by a quick baking. For device fabrication, this step is repeated three times; then, a PPA (polyphthalaldehyde, Sigma) solution (1.3 wt% in anisole) is spin-coated on PMGI (conditions: 2000 r.p.m. at 500 r.p.m. for 4 s and then 3,000 r.p.m. at 500 r.p.m. for 35 s) followed by a quick baking. With these conditions, a 20-nm-thick PPA film on top of a 210-nm-thick PMGI film is deposited on the sample surface. Patterning of PPA is performed using a commercial NanoFrazor t-SPL tool (SwissLitho AG), and FET geometries, e.g., the four-probe configuration, have been defined precisely. For the chemical etching of PMGI after patterning, samples are immersed in a solution of TMAH in deionized water (tetramethylammonium hydroxide AZ726 MIF, MicroChemicals) (0.17 mol l$^{-1}$) for 400 s, then rinsed with deionized water (30 s) and IPrOH (30 s), and finally dried with $N_2$. Metal deposition of Cr/Au (10 nm/20 nm) is performed using an AJA Orion 8E e-beam evaporator (pressure ~10$^{-8}$ torr, evaporation rate: 1 Å s$^{-1}$). Finally, metal/resist lift-off is performed by dipping samples in Remover PG (MicroChem) for a few hours, followed by rinsing (IPrOH) and drying ($N_2$).

Electrical characterization of FETs is carried out using a parameter analyzer (Agilent 4155C) and a home-built shielded probe station working in vacuum (10$^{-4}$ torr) with six micro-manipulated probes. A LabVIEW program has been developed to perform multistep measurements automatically, e.g., transfer curve, output curves, diode rectification, and parameter sweeping. The carrier mobilities of the as-grown, tc-SPL n-doped, and p-doped FETs are shown in Supplementary Figs. 19, 20.

## Data availability

The data that support the findings of this study are available from the corresponding authors on reasonable request.

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

## Acknowledgements

This article is based upon work supported by the U.S. Department of Energy, Office of Basic Energy Sciences (grant no. DE-SC0018924). We thank Prof. Angelo Bongiorno at the CUNY College of Staten Island for guidance and relevant discussions. We thank Prof. Robert Szoszkiewicz for valuable ideas and discussions. We thank Dr. Robert Keyse at Lehigh University for assistance with TEM imaging, in particular M.D. and P.M.D. acknowledge funding from NSF, Grant EFRI 2-DARE (EFRI-1542707) for TEM work.

## Author contributions

E.R. conceived, developed, and directed the entire research project. X.Z., A.C., X.L., Z.H., E.A., F.L., T.-D.L., and V.N. performed the experimental studies. X.Z. and A.C. analyzed the data. T.C. performed the DFT calculations. M.D. and P.M.D. performed the TEM experiments, C.A. directed the XPS experiments. J.W.H. designed the environmental chamber under the direction of M.V., D.S. directed the FET measurements. W.P.K. provided the tc-SPL probes. All authors contributed to the paper writing and revisions.

## Competing interests

The authors declare no competing interests.
