## [Peer Review File · Nature Communications]

Reviewers' comments:

Reviewer #1 (Remarks to the Author):

Calò et al report defects engineering in MoS₂ flakes using thermochemical scanning probe lithography assisted with reactive gas, which can form either p- or n-type semiconductors. Understanding the formation nature of defects in 2D materials and their active control are of crucial importance for engineering different properties. Oxidation scanning probe lithography has also been reported to pattern MoS₂ transistors (APPLIED PHYSICS LETTERS 106, 103503, 2015). This application of using thermochemical scanning probe lithography for creating defects in MoS₂ is new and gives different types of defects. The authors further applied XPS and DFT to give a molecular level understanding for the nature of these defects. Overall, to the reviewer, this work presents a good approach for engineering defects in MoS₂ and controlling its electrical performance. However, the introduction and discussion are not clear to deliver the main contribution and novelty of the current work. Most importantly, although approaches like KPFM and DFT have been introduced, the current work lacks of high resolution characterization to support its conclusion about the atomic nature for the defects. The thermochemical scanning probe lithography approach for defect creation is similar to STM based method for engineering defects. The current way has advantages of higher throughput, but lacks of nanoscale resolutions. The authors need to compare this approach with STM tip induced electrochemical reaction, electron beam and laser induced defect patterning. Relevant references should be discussed. My detailed comments are listed below.

The height profile of AFM image in Figure 1b needs to be provided to compare the thickness fluctuations of these layers.

HR-TEM is suggested to image the atomic nature of the formed defects, where chemical structure can be easily revealed from the imaging contrast (Nat. Commun. 6:6293, 2015). The experiments are able to provide direct evidence for the main claims made in the XPS and DFT conclusion. Electrical measurements of n-type MoS₂ made from N₂-tc-SPL shall be shown in the current work to give the difference between patterned MoS₂ and 'natural' n-type MoS₂ (without treatment). For the diode junction in Figure 2, the reviewer believes the authors used unpatterned MoS₂ flake as the n channel. This figure can be improved by using both HCl/H₂O-tc-SPL and N₂-tc-SPL patterned junction, and a higher rectification ratio can be expected.

DFT calculation has large errors in correlating the results. The XPS result in Figure 3 does not involve the substitution of oxygen atoms.

Why HCl/H₂O/N₂ has been preferentially chosen in the experiments. Its role in the thermochemical reactions needs to be further explored. Is there any other gas to be tested to demonstrate the controllability of this approach.

The patterned density and spatial distribution of defects in MoS₂ are not discussed in the current manuscript.

Reviewer #2 (Remarks to the Author):

In this article, the Authors report the realization of differently doped MoS₂ layers with integration in a diode. One of the strengths of this approach is the experimental approach that may have potentials for better scalability with respect to other top-down methods. In my opinion this demonstration will serve as a basis for future developments along the area.

The Authors supplement their work by a joint experimental and theoretical surface science analysis, reproducing in a controllable way the main findings achieved under less-controlled but more realistic conditions. This apparently suggests the main mechanisms beyond formation of p/n character. Especially the case of p-doped samples is discussed with detail, and atomistic models are proposed on the basis of XPS measurements and ab initio DFT simulations.

The work appears to be robust and presented clearly and at a sufficient level of detail, with

extensive use of references. There appear that the discussion of the nature of defects deserves some improvement and I recommend publication upon addressing the following comments:

(1) The agreement between measured and calculated core level shifts (Fig.4e) is over-emphasized. It is clear that many model structures could be proposed, and that the Authors necessarily focus on few, and that the trend is in nice agreement, yet the computed values may double the experimental ones and no comment on this issue is provided.

(2) The link of the DOS of Fig.4a-c with the p-character is not clear. What is the reader expected to look at? The fact that some DOS extends from the VB to the Fermi or slightly above? If so, is this robust with respect to the broadening scheme taken? The nature of the states at about the Fermi energy is not commented, and could be elucidated by projected DOS/bands as well as wave function amplitudes. And in the text, "the charge redistribution shows that each S atom at the surface", "at" means "above", I presume.

(3) A much lower level of detail is provided for the structures of the n-doped samples. This is imputed to MoO₃ species, but I recommend the Author provide additional explanation on the lines of the previous section, or in case state why this cannot be given. Do the Authors implicitly assume the additional O is imputed to come from the SiO₂ substrate? What are the evidences? What is the nature of the states shown in Fig

Various:

Calculations: The cutoff of 100Ry is for the charge density, or for the wave function? What is the vacuum size in the simulation cell? How is the Brillouin zone sampled? Reporting the n x m lateral size would also be helpful. Do all calculation require spin polarization?

Last line of page 3, just before Eq.(1): Ref35 would be more direct here than Ref34.

At page 7, "(see Fig. S7 of the Supplementary Information)" should be Fig. S6.

Figure 4f currently replicates Figure 3. Units are missing in the y-axes of the DOS in Fig.4.

Supplementary Fig.S10 misses color scheme for the atoms.

Reviewer #3 (Remarks to the Author):

The authors use the thermochemical scanning probe lithography technique in presence of a reactive gas to achieve nanoscale control of the local thermal activation of defects in monolayer MoS₂. The nanopatterns "drawn" by means of the above-mentioned technique can give rise to either p- or n-type conductivity on demand, depending on the reactive gas, therefore allowing to fabricate p-n junctions with a given precision and spatial control. Doping and defects formation mechanisms are studied by means of XPS and DFT, while the work function shift was measured by KPFM.

Indeed, this work reports on quite fascinating findings with potential technological relevance but some major revisions are needed:

1. The authors report the results of the electrical characterization only when p-type regions are patterned.

In addition, they should characterize ELECTRICALLY the following cases:

a. An FET device whose active area is an only n-doped region (please, report mobility and V_{th})

b. An FET device whose active area is an only p-doped region (please, report mobility and V_{th})

In case a. and b. the authors should also report the measured mobility and threshold voltage values and correlate them with the already-measured XPS, KPFM and DFT results.

2. The I_d - V_g curves in Figure 2 (especially 2b) should be plotted in logarithmic scale on the Y axis.

As well, mobility and threshold voltage values must be calculated for devices in Figure 2a and 2b, respectively. The value of the threshold voltage should be related to doping and number of induced defects.

3. The p-n device in Figure 2b is made by thermal "drawing" of a p-type region with respect to an otherwise n-type pristine region. How would the device characteristics change in Fig 2b with changing the p-type/n-type region in the channel? The authors should try to perform such an experiment.

4. It would be preferable to have more details on reactive gas concentration in the tested cell. This piece of information would make the experiment easily reproducible in other laboratories.

Reviewer #1 (Remarks to the Author):

“Calò et. al. report defects engineering in MoS₂ flakes using thermochemical scanning probe lithography assisted with reactive gas, which can form either p- or n-type semiconductors. Understanding the formation nature of defects in 2D materials and their active control are of crucial importance for engineering different properties. Oxidation scanning probe lithography has also been reported to pattern MoS₂ transistors (APPLIED PHYSICS LETTERS 106, 103503, 2015). This application of using thermochemical scanning probe lithography for creating defects in MoS₂ is new and gives different types of defects. The authors further applied XPS and DFT to give a molecular level understanding for the nature of these defects. Overall, to the Reviewer, this work presents a good approach for engineering defects in MoS₂ and controlling its electrical performance.”

We thank the Reviewer for acknowledging the novelty of our work.

(1) *“However, the introduction and discussion are not clear to deliver the main contribution and novelty of the current work. Most importantly, although approaches like KPFM and DFT have been introduced, the current work lacks of high resolution characterization to support its conclusion about the atomic nature for the defects.”*

We thank the Reviewer for this comment. As suggested by the Reviewer, in the revised version of the manuscript, we have improved the introduction and discussion part. More importantly, we have performed a new set of experiments using high resolution scanning transmission electron microscopy (STEM) to further investigate the atomic nature of the here-described defects. The detailed results on STEM have been included in the point-to-point responses below (Reviewer #1, Point #4) and added in the revised manuscript (see new Fig. 4 in revised manuscript).

(2) *“The thermochemical scanning probe lithography approach for defect creation is similar to STM based method for engineering defects. The current way has advantages of higher throughput, but lacks of nanoscale resolutions. The authors need to compare this approach with STM tip induced electrochemical reaction, electron beam and laser induced defect patterning. Relevant references should be discussed. My detailed comments are listed below.”*

We thank the Reviewer for this comment. As suggested by the Reviewer, we have reviewed other techniques besides thermochemical scanning probe lithography (tc-SPL) which are capable to locally modify the surface of two-dimensional (2D) materials, generating defects in a controlled way, and added the relevant references.

The introduction of the revised manuscript has a new paragraph (Paragraph 2 on Page 2) with the following text: *“Various direct patterning methods have been demonstrated in literature, by using either localized electric fields, electron radiation, or laser writing. Scanning tunneling microscopy (STM) for example has been shown to be able to characterize single defects in monolayer 2D materials, and also to induce defects by means of local electrochemical reactions¹⁻⁶. Oxidation scanning probe lithography was recently*

applied to pattern insulating barriers on MoS₂ flakes⁷. Electron beam radiation is also known to generate chalcogen vacancies in 2D materials^{8,9}. Finally, laser writing has been used for also local oxidation, thinning or patterning of monolayer transition metal dichalcogenides (TMDCs)¹⁰⁻¹².”

Compared to STM and other SPL methods, tc-SPL has a higher throughput (linear processing speed is in the order of ~1.5 mm/s)¹³, while compared to laser writing tc-SPL has a very good resolution (as small as 10 nm)¹⁴⁻¹⁶. Moreover, tc-SPL does not need ultra-high vacuum, and the high operational costs associated with it. Furthermore, as demonstrated in this manuscript, tc-SPL is very versatile, and the possibility to introduce various gasses during the patterning process allows for a rich variety of chemical patterning strategies for defects/doping engineering in 2D materials, hardly achieved by using other techniques.

(3) “The height profile of AFM image in Fig. 1b needs to be provided to compare the thickness fluctuations of these layers.”

We thank the Reviewer for this suggestion. The height profile of the three-layer MoS₂ after tc-SPL p-type doping (as described in Fig. 1b) has been plotted below as Fig. R1, and included in the inset of revised Fig. 1b in the manuscript. The thickness fluctuations after the tc-SPL treatment are identified to be around 0.1 nm, corresponding to less than 5% of the total thickness of the used three-layer MoS₂ flake. The thickness of a standard monolayer MoS₂ has also been marked as a guide for the eye.

Fig. R1. Thickness fluctuations of a three-layer MoS₂ flake after tc-SPL p-type doping (black dots). The thickness of a standard monolayer MoS₂ has also been marked as a guide for the eye (red dashed curve).

(4) “HR-TEM is suggested to image the atomic nature of the formed defects, where chemical structure can be easily revealed from the imaging contrast (Nat. Com. 6:6293, 2015). The experiments are able to provide direct evidence for the main claims made in the XPS and DFT conclusion.”

We thank the Reviewer for this helpful suggestion. High-angle annular dark-field (HAADF) STEM images of MoS₂ samples were collected at Lehigh University to confirm the formation of protruding covalent S-S bonds on the surface of MoS₂ when heating MoS₂ in

HCl/H₂O atmosphere, as suggested by X-ray photoelectron spectroscopy (XPS) and density functional theory (DFT) in our manuscript (we remark that these defects were showing p-character in MoS₂). Monolayer chemical vapor deposition (CVD) grown flakes (2dlayer supplies) and multilayer exfoliated flakes from bulk MoS₂ crystals (SPI supplies) on Si/SiO₂ were used. Doping was performed under N₂ previously flown through HCl solution (2.4 N) at the exactly same condition as for the XPS measurements in the manuscript. STEM measurements were obtained using a spherical aberration-corrected JEOL 200ARM-CF with an acceleration voltage of 80 kV. Images were taken with a HAADF detector with a detection range of 54-220 mrad and 10 cm camera length while electron radiation damage was minimized by using a low electron probe current (11 pA). The atomic resolution STEM images of individual defects in monolayer CVD and exfoliated MoS₂ are shown below in Fig. R2, and added in the revised manuscript as new Fig. 4. In agreement with previous XPS and DFT results which indicated a rearrangement of sulfur (S) atoms and the formation of new protruding covalent S-S bonds on the surface, the mass contrast behavior ($\sim Z^2$ where Z = atomic number) of HAADF imaging reveals a noticeable intensity rise at a chalcogen lattice site, which cannot be seen in the untreated samples, confirming an additional S atom on top of the S-site of the MoS₂ matrix (hereafter called S₃ dopants).

Fig. R2. (a) HAADF STEM lattice image of a CVD-grown monolayer MoS₂ flake exposed to p-character treatment, as described in the main text. As a result of the mass contrast behavior ($\sim Z^2$ where Z = atomic number) of HAADF imaging, Mo atoms ($Z = 42$) appear as bright spots in a 2H (trigonal prismatic) coordination with ordinary S ($Z = 32$) lattice sites (labeled as S₂). We observe the presence of dopants (outlined in green and yellow) that are not seen in pristine, untreated samples. (b) High magnification STEM image of an individual dopant from (a) (outlined in green) in which a noticeable increase in contrast is detected at a chalcogen lattice site. (c) Intensity profile across the dopant site reveals not only the Mo and ordinary S atoms (S₂), but more importantly an intensity rise that is consistent with an additional S atom on top of the S-site, thus suggesting a formation of new protruding covalent S-S bonds (S₃ dopant). (d-f) Numerous dopants observed across different CVD MoS₂ flakes demonstrate similar structural and contrast

features. (g-h) HAADF STEM lattice image of an exfoliated monolayer MoS₂ flake exposed to p-character treatment, showing S₃ dopants.

(5) *“Electrical measurements of n-type MoS₂ made from N₂-tc-SPL shall be shown in the current work to give the difference between patterned MoS₂ and ‘natural’ n-type MoS₂ (without treatment).”*

We thank the Reviewer for the suggestion, which allows us to clarify the effects of the different tc-SPL doping strategies on the performance of field-effect transistor (FET) devices. We fabricated a FET and performed N₂-tc-SPL n-type doping on the entire channel (inset of Fig. R5). The electrical characterization was performed before and after the N₂-tc-SPL n-type doping. All the details related to this n-type FET and also related to more devices which have been fabricated following the suggestions of Reviewer #3, are reported below here in the answer to Reviewer #3 Point #1. These new results are also reported as new Fig. 6 in the revised manuscript and Supplementary Information (SI) Fig. S12 – Fig. S17.

(6) *“For the diode junction in Fig. 2, the Reviewer believes the authors used unpatterned MoS₂ flake as the n channel. This Figure can be improved by using both HCl/H₂O-tc-SPL and N₂-tc-SPL patterned junction, and a higher rectification ratio can be expected.”*

We thank the Reviewer for this suggestion. Yes, in the original manuscript, the n channel was a non-patterned MoS₂ flake. As suggested by the Reviewer, to improve the diode performance, we have firstly n-type N₂-tc-SPL doped the full channel of a FET and then p-type HCl/H₂O-tc-SPL doped only half channel of the same FET, obtaining a lateral p-n like junction with both p-type and n-type regions fabricated by tc-SPL. As stated above, we refer to the answer to Reviewer #3 Point #3. The new results are also reported as new Fig. 6 in the revised manuscript and Fig. S18-S19 in SI.

(7) *“DFT calculation has large errors in correlating the results. The XPS result in Figure 3 does not involve the substitution of oxygen atoms.”*

We believe that there has been a misunderstanding on this point. In our DFT calculations, we do not consider any oxygen substitution. In old Fig. 4 of the manuscript, in order to differentiate between various S atoms, we used a different color scheme for the different types of S-atoms. In particular, an S atom under an S vacancy is represented by an orange ball, the S atoms of the MoS₂ matrix which are covalently bound to an additional S atom are colored in blue, and the additional protruding S atoms are colored in green. All other atoms in this Figure are either ordinary S atoms in their “ideal” lattice sites (colored in yellow) or Mo atoms (colored in purple). No oxygen atoms are presented in the DFT results, and no oxygen is shown in the XPS results of old Fig. 3 of the manuscript.

(8) *“Why HCl/H₂O/N₂ has be preferentially chosen in the experiments. Its role in the thermochemical reactions needs to be further explored. Is there any other gas to be tested to demonstrate the controllability of this approach.”*

Different experiments to achieve tc-SPL doping of MoS₂ were initially performed in a variety of gasses, including ambient condition, a humid environment, and an environment rich in N₂. We found that mild heating in N₂ produces modified MoS₂ areas with n-character. However, we were not able to achieve stable and reproducible p-character, moreover, we observed some possible growth of MoO₃. Considering recent results in literature regarding the doping of MoS₂ with Chlorine (Cl), we decided to consider Cl as a gas for the tc-SPL environment¹⁷. Furthermore, HCl/H₂O is a relatively safe reactive gas, and it can be easily and safely flown into our doping chamber. More importantly, there are various studies on Cl etching effects on MoS₂, proving that Cl-radical adsorption partly breaks Mo-S bonds¹⁸. Similarly to the Cl-radical adsorption, our findings suggest that when heating MoS₂ in HCl/H₂O environment, the surface S atoms can be rearranged to produce S-vacancies and protruding S-S covalent bonds, which have been here related to an induced p-character in MoS₂. While certainly other gasses, including more dangerous gasses, may be used in the future to control the doping of MoS₂, the investigation of other chemical reactions and gasses are beyond the scope of this manuscript.

(9) The patterned density and spatial distribution of defects in MoS₂ are not discussed in the current manuscript.

We thank the Reviewer for bringing this point to our attention. To improve our understanding of defect characteristics at the atomic level, MoS₂ flakes were imaged using aberration-corrected STEM as described above. The mass contrast characteristic of HAADF imaging provides atomic resolution insights into the density and spatial distribution of S₃ defects, i.e. the S-S protruding surface bond, in the doped MoS₂ samples. By averaging over multiple STEM images, we obtain a S₃ defect density of roughly 2-4% of the total chalcogen sites, which corresponds to approximately 2.4×10^{13} to 4.7×10^{13} defects/cm². This information is now reported in the revised manuscript (Paragraph 3 on Page 6).

Reviewer #2 (Remarks to the Author):

“In this article, the Authors report the realization of differently doped MoS₂ layers with integration in a diode. One of the strengths of this approach is the experimental approach that may have potentials for better scalability with respect to other top-down methods. In my opinion this demonstration will serve as a basis for future developments along the area. The Authors supplement their work by a joint experimental and theoretical surface science analysis, reproducing in a controllable way the main findings achieved under less-controlled but more realistic conditions. This apparently suggests the main mechanisms beyond formation of p/n character. Especially the case of p-type samples is discussed with detail, and atomistic models are proposed on the basis of XPS measurements and ab initio DFT simulations.”

We thank the Reviewer for the comments on the quality of our manuscript.

(1) *“The work appears to be robust and presented clearly and at a sufficient level of detail, with extensive use of references. There appear that the discussion of the nature of defects deserves some improvement and I recommend publication upon addressing the following comments: The agreement between measured and calculated core level shifts (Fig.4e) is overemphasized. It is clear that many model structures could be proposed, and that the Authors necessarily focus on few, and that the trend is in nice agreement, yet the computed values may double the experimental ones and no comment on this issue is provided.*

We thank the Reviewer for this comment. In our DFT calculations, indeed various models have been proposed and calculated for monolayer MoS₂. On the other hand, XPS measurements have been performed on multi-layer MoS₂. Therefore, considering the limits of the simulations and the differences between ideal MoS₂ monolayers and realistic exfoliated multilayers structures, we expect a quantitative agreement with a large error. This note has been added into the revised manuscript (Paragraph 2 on Page 6).

(2.1) *“The link of the DOS of Fig.4a-c with the p-character is not clear. What is the reader expected to look at? The fact that some DOS extends from the VB to the Fermi or slightly above? If so, is this robust with respect to the broadening scheme taken?”*

The density of states (DOS) in old Fig. 4a-c shows that the Fermi level has been shifted towards the valence band maximum (VBM) or even across the VBM. In our experiments, ‘p-character’ means that the measured work function becomes larger, corresponding to the shift of Fermi level towards to the VBM. Therefore, the DOS results are consistent with the experimental results. To improve the readability, we have added arrows to indicate the position of VBM with respect to the Fermi level in old Fig. 4a-c (now as new Fig. 3a-c).

Yes, this result is robust, we tried different values of degauss. The shape of the DOS may change, but the Fermi level is always close to or cross the VBM. The calculated DOS of MoS₂ with one S vacancy and one S add-atom (see old Fig. 4a) under different broadening values are given in the following Fig. R3. It clearly shows that the broadening values have little impact on the relative locations of the Fermi level.

Fig. R3. The density of states of MoS₂ with one S vacancy and one S add-atom on surface with degauss equal to 0.001 Ry (a), 0.005 Ry (b) and 0.01 Ry (c).

(2.2) *The nature of the states at about the Fermi energy is not commented, and could be elucidated by projected DOS/bands as well as wave function amplitudes. And in the text, "the charge redistribution shows that each S atom at the surface", "at" means "above", I presume.*

We thank the Reviewer for the suggestions. As shown in Fig. R4, the projected DOS of MoS₂ with one ordinary S atom (S2, yellow), one S atom underneath a S-vacancy (S1, orange), and one added atom (S3, green) covalently bonded to a S atom on the surface (S4, blue) are calculated. It shows that the 3p states of S3 atoms mainly contribute the p-type character of MoS₂. These new results are also added in the revised manuscript (Paragraph 2 on Page 6) and in the revised SI (Fig. S11).

Yes, we corrected "at" with "above".

Fig. R4. (a) The spin-polarized projected DOS of specific atoms (Mo1, Mo2, S1, S2, S3, S4) in MoS₂. The side view (b) and top view (c) of the MoS₂ atomic structures. The S atom (S1) underneath a S-vacancy, the ordinary S atom (S2), the added atom (S3) covalently bonded to a S atom on the surface (S4) are colored in orange, yellow, green and blue, respectively. Mo1 is a Mo atom bonding to S4 and Mo2 is a Mo atom bonding to S1. The projected DOS proves that S3 atoms mainly contribute to the p-type character of MoS₂ as observed in the experiments.

(3) *“A much lower level of detail is provided for the structures of the n-type samples. This is imputed to MoO₃ species, but I recommend the Author provide additional explanation on the lines of the previous section, or in case state why this cannot be given. Do the Authors implicitly assume the additional O is imputed to come from the SiO₂ substrate? What are the evidences? What is the nature of the states shown in Fig ”*

KPFM measurements indicate n-doping when tc-SPL is performed at mild temperatures in Nitrogen atmosphere; the corresponding XPS spectra for samples treated similarly, indicate both an increase of S-vacancies and an increase in the content of MoO_x. While oxygen incorporation in two-dimensional MoS₂ has been linked to p-type conduction¹⁹, S vacancies in MoS₂ have been indicated as a source of n-doping in MoS₂ by several studies²⁰. However, recently, two independent STM investigations revealed a slow oxygen-substitution reaction, during which individual sulfur atoms are replaced one by one by oxygen, giving rise to solid-solution-type 2D MoS_{2-x}O_x^{1,21}. One of these studies showed that this process, obtained either in air at room temperature, or at 400 K, gives rise to MoS_{2-x}O_x samples with n-character, however they have not been able to determine whether this n-character was due to the oxygen substitutions, since the same n-character was already present in the pristine MoS₂ samples, therefore reaching no clear conclusions^{1,21}. The authors of these studies also suggest that the common chalcogen defects usually observed in the described 2D-TMD semiconductors, are indeed oxygen substitutional defects, rather than vacancies. In particular, the authors of Ref [21] indicate that substitutional O can be incorporated in MoS₂ also while annealing in vacuum, because previously adsorbed oxygen molecules on vacancy sites could split and leave the O behind. Our MoS₂ samples are flowed with N₂ before starting the tc-SPL experiments, and the surface is imaged by AFM in contact mode, likely removing H₂O, O₂, or CO₂ adsorbed on the surface. During the tc-SPL process the cell is filled with N₂, and we therefore argue that we create S vacancies, which give rise to a n-character to the sample. The samples studied by XPS have also been “bulk” heated in N₂, and indeed they show S-vacancies, however they have not been imaged by AFM before annealing, and they also present formation of MoS_{2-x}O_x. We remind that XPS has a very poor spatial resolution, limited to above 50 μm. Likely, there are several competing phenomena when heating MoS₂ in N₂ or vacuum, namely (i) S-vacancies are created, (ii) more oxygen is incorporated in the S-vacancies increasing the amount of O in the MoS_{2-x}O_x sample, and (iii) MoO₃ is created. Process (i) is responsible for the n-character, while processes (ii) and (ii) are possibly inducing p-character, however no clear conclusions are available in literature and in our studies. It is not possible at this stage to know exactly what is the exact atomistic origin of the observed n-character, since too many effects are playing a role and it is beyond the scope of this manuscript to reveal the exact origin of this mechanism, so extensively studied in literature. However, we can conclude that the n-character is somehow related to the formation of S-vacancies in an inert atmosphere, and that after longer exposure to air the n-character disappears, possibly due to the formation of oxygen substitutional defects.

In the revised version of our manuscript we have added this discussion and the references in the revised manuscript (Paragraph 3 on Page 7).

(4) Various:

"Calculations: The cutoff of 100Ry is for the charge density, or for the wave function? What is the vacuum size in the simulation cell? How is the Brillouin zone sampled? Reporting the $n \times m$ lateral size would also be helpful. Do all calculation require spin polarization?"

Here, the norm conserving pseudopotentials are applied to describe electron-ion interactions, and an energy cutoff of 100 Ry is applied to get accurate charge density. In all simulations, 16.27 Å×16.91 Å supercells are applied to exclude the interaction between defects and their images, and a 15 Å vacuum size along the z direction is adopted to remove MoS₂ layer interaction with its images. Spin polarized calculations are carried out in all these calculations. Because spin-up and spin-down states are almost equal in non-magnetic materials, we only presented spin-up states in the old Fig. 4 of the manuscript. The corresponding Brillouin zone is sampled in a 2×2×1 mesh in the supercell optimizations. For DOS simulations, a 4×4×1 mesh is used.

This information has been added in the revised manuscript (see Methods, DFT calculations on Page 12).

"Last line of page 3, just before Eq.(1): Ref35 would be more direct here than Ref34. "

In the revised manuscript, Ref 34 has been replaced by Ref 35 in the last line of old Page 3 (now Page 4 in the revised manuscript).

"At page 7, "(see Fig. S7 of the Supplementary Information)" should be Fig. S6. "

Thanks. This has been corrected (last Paragraph on Page 7).

"Fig. 4f currently replicates Fig. 3. Units are missing in the y-axes of the DOS in Fig.4. "

Yes, the Reviewer is correct. Fig. 4f is redundant. In the revised version we have deleted old Fig. 4f and re-organized this Figure as new Fig. 3. The units of the y-axes of the DOS in old Fig. 4 are states/eV, and have been added in the revised manuscript.

"Supplementary Fig.S10 misses color scheme for the atoms. "

The color scheme for the atoms has been added in revised Fig. S10 in SI.

Reviewer #3 (Remarks to the Author):

“The authors use the thermochemical scanning probe lithography technique in presence of a reactive gas to achieve nanoscale control of the local thermal activation of defects in monolayer MoS₂. The nanopatterns “drawn” by means of the above-mentioned technique can give rise to either p- or n-type conductivity on demand, depending on the reactive gas, therefore allowing to fabricate p-n junctions with a given precision and spatial control. Doping and defects formation mechanisms are studied by means of XPS and DFT, while the work function shift was measured by KPFM.

Indeed, this work reports on quite fascinating findings with potential technological relevance but some major revisions are needed.”

(1). *“The authors report the results of the electrical characterization only when p-type regions are patterned. In addition, they should characterize ELECTRICALLY the following cases:*

- a. An FET device whose active area is an only n-type region (please, report mobility and V_{th})*
- b. An FET device whose active area is an only p-type region (please, report mobility and V_{th})*

In case a. and b. the authors should also report the measured mobility and threshold voltage values and correlate them with the already-measured XPS, KPFM and DFT results.?”

We thank the Reviewer for this advice. As suggested by the Reviewer, for the revised manuscript, we fabricated new FETs, and doped their active areas as an only n-type region and an only p-type region, respectively, using the tc-SPL method. Both types of devices have been characterized electrically before and after the tc-SPL doping. The variations of carrier mobility (μ) and threshold voltage (V_{th}) before and after tc-SPL have been reported, and correlated with the XPS, KPFM and DFT results, respectively. The details are described below here and reported in the revised manuscript.

a. FET whose active area is an only n-type region

We have fabricated a four-probe FET on CVD monolayer MoS₂. The active area of this FET has then been fully n-type doped by using tc-SPL in N₂, as described in the main text. The electrical characterizations have been performed both before and after tc-SPL n-type doping, labeled as as-grown FET and N₂-tc-SPL n-type FET, respectively.

The transfer curves before and after n-type doping have been plotted in Fig. R5 in black (as-grown) and red (N₂-tc-SPL), respectively. The threshold voltage of -32 V has been extracted for the as-grown FET before tc-SPL n-type doping, by using both the constant current method in logarithmic scale (left axis) and the linear extrapolation method in linear scale (right axis)²². After the tc-SPL n-type doping of the active region, the threshold voltage shifted from -32 V to -52 V, confirming an induced n-type behavior.

Fig. R5. Transfer curves of the FET before (black curves) and after N_2 -tc-SPL n-type doping (red curves). Inset: schematic figure showing the FET with the whole channel being n-type doped by N_2 -tc-SPL.

The carrier mobility is also extracted before and after the n-type tc-SPL via the four-probe geometry^{23,24}, and it is shown in Fig. R6.

Fig. R6. Carrier mobility of the FET before (black curve) and after (red curve) N_2 -tc-SPL n-type doping of the active area.

To correlate the electrical FET measurements with KPFM results, we use the following equation for approximating the change in the work function ($\Delta\Phi_{\text{FET}}$) of a non-degenerate semiconductor as a function of its carrier density²⁵:

$$\Delta\Phi_{\text{FET}} = \Phi_{\text{n-type}} - \Phi_{\text{as-grown}} = \ln\left(\frac{n_{\text{as-grown}}}{n_{\text{n-type}}}\right) \quad (1)$$

where $\Phi_{\text{as-grown}}$, and $\Phi_{\text{n-type}}$ are the work functions of the as-grown and n-doped MoS₂, respectively; and $n_{\text{as-grown}}$ and $n_{\text{n-type}}$ are the gate bias dependent carrier densities of the as-grown and n-doped MoS₂, extracted from the FET electrical measurements.

In particular, we approximate the bias-dependent carrier density in the channel region of the MoS₂ FET in strong inversion (i.e. at gate bias beyond the threshold voltage) using²⁶:

$$n = C_{\text{ox}}(V_g - V_{\text{th}}) \quad (2)$$

where C_{ox} is the gate capacitance, V_g is the back-gate voltage, and V_{th} is the threshold voltage. From equation (1) and using the bias-dependent carrier density before and after the n-doping (see Fig. R7), we estimated a work function shift ($\Delta\Phi_{\text{FET}}$) of -12 meV at $V_g = 0$ V, as shown in Fig. R8.

Fig. R7. The carrier density before (black curve) and after (red curve) the n-type, and the ratio (green curve) of $\frac{n_{\text{as-grown}}}{n_{\text{n-type}}}$.

Next, we also performed KPFM measurements on this FET before and after the n-type doping. From these measurements, we obtain a change in work function of $\Delta\Phi_{\text{KPFM}} = \Phi_{\text{n-type}} - \Phi_{\text{as-grown}} = -18 \pm 5$ meV, in agreement with the data reported in Fig. 5a of the main manuscript, when the temperature of the tc-SPL heater is 1000 K. Note that the substrate is grounded during the KPFM measurements. We recall that the FET work function shift ($\Delta\Phi_{\text{FET}}$) at $V_g = 0$ V extracted from the electrical measurements for this n-type character FET (Fig. R8) is -12 meV, also in good agreement with the KPFM data. However, we note that after several experiments the level of n-type is unstable in air, likely due to the filling of the S-vacancies with substitutional Oxygen, as reported in recent publications^{1,21} and discussed at the Point #3 of Reviewer #2. Furthermore, we also discovered that the n-type doping is dramatically influenced by the state of the as-grown MoS₂ sample, e.g. by the number of vacancies present at the time of the doping procedure, which are strongly dependent on the local humidity. On the other hand, we noted that the p-type doping is much more reproducible and stable.

Fig. R8. Plot of the FET work function shift between as-grown MoS₂ and tc-SPL n-type doped MoS₂, extracted from the electrical measurements. The work function at $V_g = 0$ V has been marked for direct comparison with KPFM results.

Regarding the comparison of these values with the XPS results, we remark that the work function is the energy difference between the vacuum level and Fermi energy, while the highest energy level of the valence band (VBM) is measured during the XPS experiments (VBM_{XPS}) as the energy difference, considered positive, between the Fermi energy and VBM. In particular, for n-doping we measured $\Delta\Phi = \Phi_{n\text{-type}} - \Phi_{\text{as-grown}} = -18$ meV and $\Delta VBM_{XPS} = VBM_{XPS}(n\text{-type}) - VBM_{XPS}(\text{as-grown}) = +250$ meV. These two energy shifts are in agreement since an upward shift in the Fermi energy corresponds to a decrease in the work function and a corresponding increase in VBM_{XPS} , however, the two shifts are not the same in absolute value because during the n-doping process both the VBM and the Fermi level change.

These figures and measurements of tc-SPL n-type doped FET are now reported in the new Fig. 6 in the revised manuscript, Fig. S12 – Fig. S14 and Section 7 in the SI.

b. FET whose active area is an only p-type region

Similarly, to the case of the n-type FET, we fabricated another four-probe FET on CVD monolayer MoS₂. The electrical characterizations have been performed both before and after tc-SPL p-type doping. The transfer curves, plotted in black (as-grown) and in blue (p-type) are shown in Fig. R9. The threshold voltage is found to shift from -32 V (as-grown) to -1 V after tc-SPL p-type doping of the active area. This positive shift of V_{th} is consistent with the p-type doping of MoS₂.

Fig. R9. Transfer curves of the FET before (black curves) and after (blue curves) tc-SPL p-type doping. Inset: schematic figure showing the FET with the whole channel being p-type doped by tc-SPL.

The carrier mobility before and after the p-type doping has been calculated and shown in Fig. R10.

Fig. R10. Carrier mobility of the FET before (black curve) and after (blue curve) tc-SPL p-type doping of the active area.

We then study the work function shift in the channel region of the MoS₂ device before and after the p-doping. To do so, we use the same analysis applied to the n-type devices earlier. The bias-dependent carrier density before ($n_{\text{as-grown}}$) and after ($n_{\text{p-type}}$) the p-type doping, calculated according to equation (2), and the carrier density ratio $\frac{n_{\text{as-grown}}}{n_{\text{p-type}}}$ are plotted in Fig. R11. Therefore, the FET work function shift ($\Delta\Phi_{\text{FET}} = \Phi_{\text{p-type}} - \Phi_{\text{as-grown}}$) associated

with the p-type doping can be calculated by using equation (1), and is plotted as a function of the gate voltage in Fig. R12. At $V_g = 0$, the work function shift estimated from the FET data ($\Delta\Phi_{\text{FET}}$) is found to be +90 meV, indicating a downshift of the Fermi level (p-character).

Fig. R11. The carrier density before (black curve) and after (blue curve) the p-type doping, and their ratio (green curve) of $\frac{n_{\text{as-grown}}}{n_{\text{p-type}}}$.

Fig. R12. Plot of the FET work function shift between as-grown MoS₂ and tc-SPL p-type doped MoS₂, extracted from the electrical measurements. The work function at $V_g = 0$ V has been marked for direct comparison with KPFM results.

Furthermore, we performed KPFM measurements of the FET before and after tc-SPL p-type doping. The work function variation from the KPFM contrast has been identified to

be $(+90 \pm 6)$ meV. We recall that the FET work function shift ($\Delta\Phi_{\text{FET}}$) at $V_g = 0$ V extracted from the electrical measurements for this doped FET (Fig. R12) is +90 meV, in good agreement with the KPFM data. Regarding the comparison of these values with the XPS and DFT results, we obtained the following. As shown in Fig. R13 blow, first, we remark that the work function (Φ) is the energy difference between the vacuum level and Fermi energy, while the VBM measured during the XPS experiments (VBM_{XPS}) is the energy difference, considered positive, between the Fermi energy, E_{FE} , and highest energy level of the valence band; therefore, a downward shift in the Fermi energy corresponds to an increase in the work function and a corresponding decrease in VBM_{XPS} . On the other hand, the DFT simulations calculate the highest energy level of the valence band from the vacuum level, E_{vac} , (which is assumed to be at zero energy), and we call this value VBM_{DFT} . Therefore, the different energy shifts measured in KPFM/FET, XPS, and DFT are related by the following relationship (see also as Fig. R13):

$$\text{VBM}_{\text{DFT}} = E_{\text{vac}} - \text{VBM} = (E_{\text{vac}} - E_{\text{FE}}) + (E_{\text{FE}} - \text{VBM}) = \Phi + \text{VBM}_{\text{XPS}}$$

Now considering the differences between the energy shifts for the as grown samples compared to the tc-SPL p-doped samples, we obtain:

$$\Delta\text{VBM}_{\text{DFT}} = \Delta\Phi + \Delta\text{VBM}_{\text{XPS}}$$

When introducing the results obtained in our work, we have $\Delta\Phi = (+90 \pm 6)$ meV as discussed above here, and $\Delta\text{VBM}_{\text{XPS}} = \text{VBM}_{\text{XPS}}(\text{p-type}) - \text{VBM}_{\text{XPS}}(\text{as-grown}) = -300$ meV as obtained from new Fig. 2b (old Fig. 3b) of revised manuscript. Regarding the DFT simulations, in our manuscript we have described three structure models, MoS_2 with one S vacancy and one surface S-S protruding bond (1.6% defect density), MoS_2 with one S vacancy and two surface S-S protruding bond (3.3% defect density); and MoS_2 with one S vacancy and three surface S-S protruding bond (5% defect density). If compared with a pristine monolayer MoS_2 we obtain, respectively for these structures, the following shifts: $\Delta\text{VBM}_{\text{DFT}} = \text{VBM}_{\text{DFT}}(\text{p-type}) - \text{VBM}_{\text{DFT}}(\text{as-grown}) = -150$ meV, -360 meV, and -550 meV. Since XPS elemental analysis shows a S/Mo ratio of 2.005, which remains constant throughout the doping procedure, we conclude that the majority of the defects produced during the p-doping process consists of one S vacancy and one surface S-S protruding bond, as also observed in the STEM experiments. We therefore obtain:

$$\Delta\text{VBM}_{\text{DFT}} = -150 \text{ meV vs. } -210 \text{ meV} = \Delta\Phi + \Delta\text{VBM}_{\text{XPS}},$$

which we consider a good agreement, considering the limitations of this comparison.

These figures and measurements of a tc-SPL p-type doped FET are now reported in the new Fig. 6 in the revised manuscript, Fig. S15 – Fig. S17, and Section 7 in the SI.

Fig. R13. Schematic figure showing the relation of work function (Φ) measured by KPFM and FET, VBM measured by XPS (VBM_{XPS}) and the VBM_{DFT} defined in DFT calculations. The work function shift ($\Delta\Phi$), the $\Delta\text{VBM}_{\text{XPS}}$ measured in XPS, and the $\Delta\text{VBM}_{\text{DFT}}$ calculated in DFT have been listed, respectively.

(2). “The I_d - V_g curves in Fig. 2 (especially 2b) should be plotted in logarithmic scale on the Y axis. As well, mobility and threshold voltage values must be calculated for devices in Fig. 2a and 2b, respectively. The value of the threshold voltage should be related to doping and number of induced defects.”

We thank the Reviewer for this suggestion.

As suggested by the Reviewer in Point #1 (discussed above) and Point #3 (discussed below), we have updated the manuscript with new FET devices, including a fully n-type doped FET (discussed in Point #1), a fully p-type dope FET (discussed in Point #1) and a lateral p-n like junction where both sides of the junction have been doped by tc-SPL (will be discussed in Point #3 below).

As a result, the old Fig. 2 of the manuscript has been revised. The results of the fully n-type doped FET, fully p-type doped FET and the lateral p-n like junction are now added as new Fig. 6 in the revised manuscript and Fig. S12 – Fig. S19 in SI.

(3). “The p-n device in Fig. 2b is made by thermal “drawing” of a p-type region with respect to an otherwise n-type pristine region. How would the device characteristics change in Fig 2b with changing the p-type/n-type region in the channel? The authors should try to perform such an experiment.”

We thank the Reviewer for this suggestion. As suggested by the Reviewer, the p-n junction formation begins with the n-doping of the entire channel region of a FET by N_2 -tc-SPL. We then converted the doping type in one half of the channel region to p-doping by HCl/H_2 -tc-SPL, obtaining a lateral p-n like junction with both p-type and n-type regions fabricated by tc-SPL.

The output characteristic curves of the FET after the initial n-type doping of its channel region confirmed the absence of a diode-like characteristics for the entire measured range of the gate voltage. This is evident from the symmetric output characteristics of the FET (see Fig. R14a-b). After the p-doping, however, the device shows a gate-bias-dependent rectification behavior (see Fig. R14c-d). For the fully N₂-tc-SPL n-type doped FET (Fig. R14 a-b), the output curves are symmetric at both small ($V_g = -30V$) and large ($V_g = 40V$) back gating, indicating that no diode has formed, thus showing a uniform N₂-tc-SPL n-type doping of the channel and high quality of the metal contacts. However, after the HCl/H₂O-tc-SPL half channel p-type doping, the output curves of the lateral p-n like junction (Fig. R14 c-d) show prominent rectification at small back gating ($V_g = -30V$), and become symmetric again at large back gating ($V_g = 40V$).

The rectification ratio, defined as the ratio of forward current (ON) over reverse current (OFF), has been characterized as a function of back gating, as shown in Fig. R15. For a fully N₂-tc-SPL n-type doped FET, the rectification ratio stays close to 1 at all V_g , indicating that no rectification has been observed due to the uniform N₂-tc-SPL n-type channel and the high-quality metal contacts. On the contrary, the rectification ratio of the lateral p-n junction increases with back gating and reaches a maximum at $V_g = -32 V$. As V_g increases above -32 V, the rectification ratio reduces back to 1. The transition from a diode-like behavior at low gate bias to a unipolar device characteristic at high gate bias (see Fig. R14 - R15) is due to the gradual increase in the density of gate-bias-induced electron charge carriers relative to the p-type carriers induced by tc-SPL. This gate-bias-dependent characteristic is consistent with the previous report on MoS₂ lateral p-n junctions²⁷.

To achieve the optimal rectification ratio of the lateral p-n like junction, we have fixed the back gating at -32V, and a rectification ratio of over 10^4 has been observed as shown in Fig. R16. The strong rectification behavior of the lateral p-n junction provides further evidence for the ability of tc-SPL technique in altering the doping type and doping level at precise locations of a MoS₂ device *on demand*.

Fig. R14. (a-b) Output curves of the fully N_2 -tc-SPL n-type doped FET at small ($V_g = -30$ V) and large ($V_g = 40$ V) back gating. (c-d) Output curves of the lateral p-n like junction at small ($V_g = -30$ V) and large ($V_g = 40$ V) back gating.

Fig. R15. The rectification ratio as a function of V_g for the fully N_2 -tc-SPL n-type doped FET (data in red) and the lateral p-n like junction (data in black).

Fig. R16. Output curve collected at $V_g = -32$ V with extended drain-source voltage. A maximum rectification ratio $> 10^4$ has been achieved. Inset: KPFM images of the FET channel before (top panel) and after (bottom panel) the formation of the lateral p-n junction. Both the p-type half channel (blue dashed line) and the n-type half channel (red dashed line) are realized by using the tc-SPL method.

Inset of Fig. R16 shows the KPFM image of the final device structure, indicating the formation of a lateral p-n junction. The KPFM image of an as-grown MoS_2 is also shown in the inset of Fig. R16 (top panel). By using the KPFM of the as-grown MoS_2 as a reference, it is possible to identify visually the p-doped and n-doped regions of the lateral p-n junction, where the darker CPD area (marked with a blue dashed box) corresponds to the p-doped region and the brighter CPD area (marked in red dashed box) corresponds to the n-doped region. Therefore, it is evident that both n-type and p-type half channels are realized by using the tc-SPL method. As a result, the rectification ratio has been greatly improved.

The results of the improved lateral p-n junction have been added as new Fig. 6 in the revised manuscript and Fig. S18-S19 in SI.

(4). *“It would be preferable to have more details on reactive gas concentration in the tested cell. This piece of information would make the experiment easily reproducible in other laboratories.”*

We thank the Reviewer for this suggestion. HCl gas from a 2.4 M solution is passed through a NaOH solution for 20 minutes. The change in pH of the solution is monitored to calculate the HCl gas concentration, which is 74 micromoles per liter of gas. We have added this part into revised Methods section (tc-SPL nanopatterning) in the revised manuscript (Paragraph 1 Page 11).

References

- 1 Barja, S. *et al.* Identifying substitutional oxygen as a prolific point defect in monolayer transition metal dichalcogenides. *Nature Communications* **10**, doi:10.1038/s41467-019-11342-2 (2019).
- 2 Schuler, B. *et al.* How Substitutional Point Defects in Two-Dimensional WS₂ Induce Charge Localization, Spin-Orbit Splitting, and Strain. *Acs Nano* **13**, 10520-10534, doi:10.1021/acsnano.9b04611 (2019).
- 3 Schuler, B. *et al.* Large Spin-Orbit Splitting of Deep In-Gap Defect States of Engineered Sulfur Vacancies in Monolayer WS₂. *Physical Review Letters* **123**, doi:10.1103/PhysRevLett.123.076801 (2019).
- 4 Lin, Z. *et al.* Defect engineering of two-dimensional transition metal dichalcogenides. *2D Materials* **3**, 022002 (2016).
- 5 Wong, D. *et al.* Characterization and manipulation of individual defects in insulating hexagonal boron nitride using scanning tunnelling microscopy. *Nature Nanotechnology* **10**, 949-U192, doi:10.1038/nnano.2015.188 (2015).
- 6 Velasco, J. *et al.* Nanoscale Control of Rewriteable Doping Patterns in Pristine Graphene/Boron Nitride Heterostructures. *Nano Letters* **16**, 1620-1625, doi:10.1021/acs.nanolett.5b04441 (2016).
- 7 Espinosa, F. M. *et al.* Direct fabrication of thin layer MoS₂ field-effect nanoscale transistors by oxidation scanning probe lithography. *Applied Physics Letters* **106**, doi:10.1063/1.4914349 (2015).
- 8 Komsa, H. P. *et al.* Two-Dimensional Transition Metal Dichalcogenides under Electron Irradiation: Defect Production and Doping. *Physical Review Letters* **109**, doi:10.1103/PhysRevLett.109.035503 (2012).
- 9 Komsa, H. P., Kurasch, S., Lehtinen, O., Kaiser, U. & Krasheninnikov, A. V. From point to extended defects in two-dimensional MoS₂: Evolution of atomic structure under electron irradiation. *Physical Review B* **88**, doi:10.1103/PhysRevB.88.035301 (2013).
- 10 Li, H. *et al.* Mechanical Exfoliation and Characterization of Single- and Few-Layer Nanosheets of WSe₂, TaS₂, and TaSe₂. *Small* **9**, 1974-1981, doi:10.1002/smll.201202919 (2013).
- 11 Castellanos-Gomez, A. *et al.* Laser-Thinning of MoS₂: On Demand Generation of a Single-Layer Semiconductor. *Nano Letters* **12**, 3187-3192, doi:10.1021/nl301164v (2012).
- 12 Seo, S. Y. *et al.* Writing monolithic integrated circuits on a two-dimensional semiconductor with a scanning light probe. *Nature Electronics* **1**, 512-517, doi:10.1038/s41928-018-0129-6 (2018).
- 13 Garcia, R., Knoll, A. W. & Riedo, E. Advanced scanning probe lithography. *Nat Nanotechnol* **9**, 577-587 (2014).
- 14 Szoszkiewicz, R. *et al.* High-speed, sub-15 nm feature size thermochemical nanolithography. *Nano Lett* **7**, 1064-1069, doi:10.1021/nl070300f (2007).
- 15 Wei, Z. Q. *et al.* Nanoscale Tunable Reduction of Graphene Oxide for Graphene Electronics. *Science* **328**, 1373-1376, doi:10.1126/science.1188119 (2010).
- 16 Liu, X. Y. *et al.* Sub-10 nm Resolution Patterning of Pockets for Enzyme Immobilization with Independent Density and Quasi-3D Topography Control. *Acs*

- Applied Materials & Interfaces* **11**, 41780-41790, doi:10.1021/acsami.9b11844 (2019).
- 17 Yang, L. M. *et al.* Chloride Molecular Doping Technique on 2D Materials: WS₂ and MoS₂. *Nano Letters* **14**, 6275-6280, doi:10.1021/nl502603d (2014).
- 18 Kim, K. S. *et al.* Atomic Layer Etching Mechanism of MoS₂ for Nanodevices. *Acs Applied Materials & Interfaces* **9**, 11967-11976, doi:10.1021/acsami.6b15886 (2017).
- 19 Neal, A. T., Pachter, R. & Mou, S. P-type conduction in two-dimensional MoS₂ via oxygen incorporation. *Applied Physics Letters* **110**, 193103, doi:10.1063/1.4983092 (2017).
- 20 Lei, S. *et al.* Surface functionalization of two-dimensional metal chalcogenides by Lewis acid-base chemistry. *Nat Nanotechnol* **11**, 465-471, doi:10.1038/nnano.2015.323 (2016).
- 21 Peto, J. *et al.* Spontaneous doping of the basal plane of MoS₂ single layers through oxygen substitution under ambient conditions. *Nature Chemistry* **10**, 1246-1251, doi:10.1038/s41557-018-0136-2 (2018).
- 22 Ortiz-Conde, A. *et al.* A review of recent MOSFET threshold voltage extraction methods. *Microelectronics Reliability* **42**, 583-596, doi:10.1016/s0026-2714(02)00027-6 (2002).
- 23 Schroder, D. K. *Semiconductor Material and Device Characterization*. 3rd edn, (John Wiley & Sons, Inc., Publication).
- 24 in *Semiconductor Material and Device Characterization* 61-125.
- 25 Kittel, C. *Introduction to Solid State Physics*. (Wiley, 2004).
- 26 Schroder, D. K. *Semiconductor Material and Device Characterizations* (Third Edition). *John Wiley & Sons, Inc., Publication* (2006).
- 27 Gao, L. *et al.* Defect-Engineered Atomically Thin MoS₂ Homogeneous Electronics for Logic Inverters. *Advanced materials* **32**, doi:10.1002/adma.201906646 (2020).

REVIEWER COMMENTS

Reviewer #1 (Remarks to the Author):

The authors have done a good job in clarifying reviewer's concerns, with significant time spent on improving the manuscript. I appreciate the author's efforts. Particularly, the new STEM results provided direct evidence to back up the claims in the manuscript. As it is natural to get n-type MoS₂, the reviewer has one remaining question regarding the reproducibility and yield of p-type MoS₂ fabricated in HCl environments by using this method. In other words, how many devices are made in this p-type doping experiments and what is the successful rate. In the manuscript (Page 8, Line 9), the authors mentioned that tc-SPL p-type doping is more reproducible and stable, but did not provide such discussions.

Reviewer #2 (Remarks to the Author):

The Authors have strengthened their work with extensive additional experiments and analysis. They appear to resolve all issues indicated on the previous submission by me and (up to my understanding) the other reviewers.

One minor point that does not impact the relevance of the paper, but that should be adjusted, concerns my comment on the link of the DFT-DOS with the p-type character [from the first report: "(2) The link of the DOS of Fig.4a-c with the p-character is not clear. What is the reader expected to look at? The fact that some DOS extends from the VB to the Fermi or slightly above? If so, is this robust with respect to the broadening scheme taken?"]

As a matter of fact, the results shown by the Authors in their reply (Fig.R3) suggest that the Fermi level is moving downwards in the valence band as broadening increases. Recall that the integral of the DOS up to E_F should be constant (it must provide the number of electrons in the system), but this situation typically arises because of the common computational practice. Indeed, one computes the self-consistent density with Methfessel-Paxton smearing (as the Authors do) and, given M-P smearing functions are non-positively defined, the Fermi level is computationally fixed at about the valence band edge. Taking symmetric and positively defined Gaussian occupations brings the Fermi level at about the middle of the gap. Plotting the DOS with positively-defined smearing functions is then usual practice, since it removes nasty negative regions in the plotted DOS. Here one also uses a smaller broadening than e.g. the 0.02 Ry used for self-consistency. By looking to the figures, this is also what the Authors do. Summarizing, this is all common practice and I don't believe the Authors should provide further details in the paper. However, no physical implications should be derived from this numerical issue, that is basically irrelevant for all computational purposes, and the text around line 235 "the Fermi level of MoS₂ is shifted down to or even below the valence band maximum" should be removed. So the information that samples are p-doped only comes from the experiments.

Reviewer #3 (Remarks to the Author):

The authors have addressed all the issues raised during the first review round in a satisfactory manner. The paper can be published in its present form.

Reviewer #1 (Remarks to the Author):

“The authors have done a good job in clarifying reviewer’s concerns, with significant time spent on improving the manuscript. I appreciate the author’s efforts. Particularly, the new STEM results provided direct evidence to back up the claims in the manuscript.

We thank the Reviewer for acknowledging the improvement of the manuscript and our efforts.

“As it is natural to get n-type MoS₂, the reviewer has one remaining question regarding the reproducibility and yield of p-type MoS₂ fabricated in HCl environments by using this method. In other words, how many devices are made in this p-type doping experiments and what is the successful rate. In the manuscript (Page 8, Line 9), the authors mentioned that tc-SPL p-type doping is more reproducible and stable, but did not provide such discussions.”

We thank the Reviewer for this suggestion.

In our tc-SPL doping system, a wide range of parameters have been studied extensively and optimized in order to achieve the most reproducible p-type (and n-type) doping of MoS₂. These parameters include: 1) the concentration of the HCl solution; 2) the gas flow rate; 3) the gas pre-flow duration; 4) the setpoint load of thermal cantilever; 5) the number of scanned lines along the slow axis (y-axis); 6) the scan rate; and 7) the temperature of the heater. We have performed over 500 p-type doping experiments to evaluate the influence of the 7 parameters on the HCl/H₂O-tc-SPL doping of MoS₂ in terms of doping level, uniformity, doping speed, flake damage and reproducibility. For example, Fig. 1c and Fig. 5a of the manuscript present the results of some of these experiments where the doping level is measured as a function of scan rate and temperature after all the other parameters have been optimized (see Methods in the manuscript).

After the optimization, we have fabricated in total 30 p-type doping devices and observed consistent work function change for 28 of these devices (yield ~ 93%). It should be noted that the successful p-type doping level of the devices depends also on the MoS₂ samples, e.g. exfoliated flakes or chemical vapor deposition (CVD) flakes.

We have added this discussion into a new section of Parameter optimization in the Methods part (Paragraph 2 on Page 11).

Reviewer #2 (Remarks to the Author):

“The Authors have strengthened their work with extensive additional experiments and analysis. They appear to resolve all issues indicated on the previous submission by me and (up to my understanding) the other reviewers.”

We thank the Reviewer for acknowledging the improvement of the manuscript.

“One minor point that does not impact the relevance of the paper, but that should be adjusted, concerns my comment on the link of the DFT-DOS with the p-type character [from the first report: “(2) The link of the DOS of Fig.4a-c with the p-character is not clear. What is the reader expected to look at? The fact that some DOS extends from the VB to the Fermi or slightly above? If so, is this robust with respect to the broadening scheme taken?”]

As a matter of fact, the results shown by the Authors in their reply (Fig.R3) suggest that the Fermi level is moving downwards in the valence band as broadening increases. Recall that the integral of the DOS up to E_F should be constant (it must provide the number of electrons in the system), but this situation typically arises because of the common computational practice. Indeed, one computes the self-consistent density with Methfessel-Paxton smearing (as the Authors do) and, given M-P smearing functions are non-positively defined, the Fermi level is computationally fixed at about the valence band edge. Taking symmetric and positively defined Gaussian occupations brings the Fermi level at about the middle of the gap.

Plotting the DOS with positively-defined smearing functions is then usual practice, since it removes nasty negative regions in the plotted DOS. Here one also uses a smaller broadening than e.g. the 0.02 Ry used for self-consistency. By looking to the figures, this is also what the Authors do.

Summarizing, this is all common practice and I don't believe the Authors should provide further details in the paper. However, no physical implications should be derived from this numerical issue, that is basically irrelevant for all computational purposes, and the text around line 235 “the Fermi level of MoS₂ is shifted down to or even below the valence band maximum” should be removed.

So the information that samples are p-doped only comes from the experiments.”

We thank the Reviewer for explaining this numerical issue again in details for us. We appreciate that the Reviewer points out that this is common practice and no further details in the paper should be provided. We agree that the relative position between valence band maximum (VBM) and fermi level will be affected by the used smearing method and broadening values.

As suggested by the Reviewer, we have removed the text “*the Fermi level of MoS₂ is shifted down to or even below the valence band maximum*”. In addition, we have updated Fig. 3 in the manuscript (see also Fig. R1 below). We have removed the position of Fermi level (E_F) and labeled only the VBM.

We have revised the corresponding paragraph in the manuscript (Paragraph 3 on Page 5) as follows:

“The DFT calculations (see Methods for details) on surface protruding S-S covalent bond models demonstrate that top sites of S atoms in the MoS₂ matrix are the most stable adsorption-positions of extra S atoms (Fig. S8 in SI). The projected density of states (PDOS) of MoS₂ with one S vacancy and an increasing number of protruding surface S

atoms are calculated and shown in Fig. 3a (one protruding surface S atom, S_V-1S_d), 3b (two protruding surface S atoms, S_V-2S_d) and 3c (three protruding surface S atoms, S_V-3S_d), where the energy levels of the states are calculated with respect to the vacuum level. It is evident that the protruding surface S atoms are the major contributors to the VBM_{DFT} shift towards higher energy levels with respect to pristine MoS_2 (black dashed line, see also Fig. S9 in SI). Furthermore, the shift of VBM_{DFT} in DFT, calculated from the vacuum level, has been correlated with XPS and KPFM results, and a good agreement has been achieved (see Section 7 in SI). Moreover, we demonstrate that Mo vacancies do not produce p-type doping in MoS_2 , see Fig. S7 in SI, and when considering also their large formation energy, we conclude that Mo vacancies have no effect in our experiments.”

Fig. R1. **DFT results.** Atom normalized spin-polarized projected density of states (PDOS) of MoS_2 with one S vacancy and one (S_V-1S_d) (a), two (S_V-2S_d) (b), and three (S_V-3S_d) (c) protruding surface S atoms chemisorbed on the surface. The energy level of the valence band maximum (VBM_{DFT}) is calculated with respect to the vacuum level and it is indicated by a blue dashed line, while the VBM of pristine MoS_2 is shown with a black dashed line and obtained from Ref. [1]. (d) The schematic figures of corresponding MoS_2 structures used in PDOS calculations. The S atom (S1) underneath a S-vacancy, the ordinary S atom (S2), and the protruding S atom (S3) covalently bonded to a S atom on the surface (S4) are colored in orange, yellow, green and blue, respectively. Mo atoms are represented by violet balls. (e) The core-level energy shifts of S 2p states with respect to ordinary S atom (S2) (0.00 eV) are indicated. (f) Histogram of the DFT results (in dark blue and dark green) for the bottom (S4) and protruding S (S3) atoms with creation of a S vacancy together with the XPS energy shifts (in light green and light blue) of the two additional doublets used for the fit.

Reviewer #3 (Remarks to the Author):

“The authors have addressed all the issues raised during the first review round in a satisfactory manner. The paper can be published in its present form.”

We thank the Reviewer for suggesting publication of the manuscript.

References:

1. Komsa, H. P. & Krasheninnikov, A. V. Native defects in bulk and monolayer MoS₂ from first principles. Physical Review B 91, 125304 (2015)

REVIEWERS' COMMENTS:

Reviewer #1 (Remarks to the Author):

The authors have addressed my concerns. I recommend the publication of this matter.

Reviewer #2 (Remarks to the Author):

The Authors have addressed all comments and in my opinion the manuscript can be published in its current form.

Reviewer #1 (Remarks to the Author):

“The authors have addressed my concerns. I recommend the publication of this matter.”

We thank the Reviewer for suggesting publication of the manuscript.

Reviewer #2 (Remarks to the Author):

“The Authors have addressed all comments and in my opinion the manuscript can be published in its current form.”

We thank the Reviewer for suggesting publication of the manuscript.